# The PRC2.1 subcomplex opposes G1 progression through regulation of CCND1 and CCND2

Adam D Longhurst[1,2], Kyle Wang[3,4], Harsha Garadi Suresh[3], Mythili Ketavarapu[5,6], Henry N Ward[7], Ian R Jones[8,9], Vivek Narayan[8], Frances V Hundley[1,2,10], Arshia Zernab Hassan[11], Charles Boone[3,4], Chad L Myers[7,10], Yin Shen[8,12,13], Vijay Ramani[5,6], Brenda J Andrews[4], David P Toczyski[1,6,13]*

[1]University of California, San Francisco, San Francisco, United States; [2]Tetrad Graduate Program, University of California, San Francisco, San Francisco, United States; [3]Department of Molecular Genetics, University of Toronto, Toronto, Canada; [4]The Donnelly Centre for Cellular and Biomolecular Research, University of Toronto, Toronto, Canada; [5]Gladstone Institute for Data Science and Biotechnology, J. David Gladstone Institutes, San Francisco, United States; [6]Department of Biochemistry and Biophysics, University of California, San Francisco, San Francisco, United States; [7]Bioinformatics and Computational Biology Graduate Program, University of Minnesota – Twin Cities Minneapolis, Minneapolis, United States; [8]Institute for Human Genetics, University of California, San Francisco, San Francisco, United States; [9]Pharmaceutical Sciences and Pharmacogenomics Graduate Program, University of California, San Francisco, San Francisco, United States; [10]Department of Cell Biology, Blavatnik Institute of Harvard Medical School, Boston, United States; [11]Department of Computer Science and Engineering, University of Minnesota – Twin Cities Minneapolis, Minneapolis, United States; [12]Department of Neurology, University of California, San Francisco, San Francisco, United States; [13]Weill Institute for Neurosciences, University of California, San Francisco, San Francisco, United States

*For correspondence:
dpt4darwin@gmail.com

Competing interest: The authors declare that no competing interests exist.

## eLife Assessment

This **valuable** study reports a chemogenetic screen for resistance and sensitivity to three cell cycle inhibitors used in the clinic: camptothecin, colchicine, and palbociclib. The screen provides a wealth of information that will be of interest to cell cycle and cancer biologists. **Convincing** evidence is provided that resistance to palbociclib can result from loss of PRC2.1 activity, which raises cyclin D levels. The effect of PRC2.1 on cyclin D is not universal across tested cell lines with the causal differences not yet understood.

**Abstract** Progression through the G1 phase of the cell cycle is the most highly regulated step in cellular division. We employed a chemogenetic approach to discover novel cellular networks that regulate cell cycle progression. This approach uncovered functional clusters of genes that altered sensitivity of cells to inhibitors of the G1/S transition. Mutation of components of the Polycomb Repressor Complex 2 rescued proliferation inhibition caused by the CDK4/6 inhibitor palbociclib, but not to inhibitors of S phase or mitosis. In addition to its core catalytic subunits, mutation of the PRC2.1 accessory protein MTF2, but not the PRC2.2 protein JARID2, rendered cells resistant to palbociclib treatment. We found that PRC2.1 (MTF2), but not PRC2.2 (JARID2), was critical for

promoting H3K27me3 deposition at CpG islands genome-wide and in promoters. This included the CpG islands in the promoter of the CDK4/6 cyclins CCND1 and CCND2, and loss of MTF2 lead to upregulation of both CCND1 and CCND2. Our results demonstrate a role for PRC2.1, but not PRC2.2, in antagonizing G1 progression in a diversity of cell linages, including chronic myeloid leukemia (CML), breast cancer, and immortalized cell lines.

## Introduction

Cellular decisions to grow and divide are made by assessing the balance of activating and inhibitory inputs that govern the transition between cell cycle phases. Regulated progression through the cell cycle is crucial for normal cellular growth and organismal development (*Ginzberg et al., 2015*; *Massagué, 2004*; *Bertoli et al., 2013*). Progression from G1 into S phase is the most highly regulated step of the cell cycle, as initiating DNA replication commits a cell to divide and is frequently mutationally activated in tumors. Cyclin-Dependent Kinase 4 (CDK4) and the related CDK6 (henceforth referred to collectively as CDK4/6) play critical roles in promoting G1 progression through phosphorylation of the retinoblastoma protein (RB1). Phosphorylation relieves RB1-mediated transcriptional repression of E2F transcription factors, which are then competent to drive transcription of genes necessary for progression into S phase (*Bertoli et al., 2013*; *Rubin et al., 2020*; *Malumbres and Barbacid, 2005*; *Bracken et al., 2004*). Because of their crucial role in regulating G1 progression, specific inhibitors targeting CDK4/6 have proven to be effective therapeutics. Palbociclib was the first FDA approved CDK4/6 inhibitor and highly efficacious in the treatment of HR+/HER2– breast cancers, followed by the structurally related molecules ribociclib and abemaciclib (*Dickler et al., 2017*; *Morrison et al., 2024*; *Turner et al., 2018*; *Im et al., 2019*; *Goel et al., 2022*). However, this classical model of G1 regulation has recently given way to a more complex model (*Cappell et al., 2018*; *Cappell et al., 2016*; *Chung et al., 2019*), underscored by the complexity of genetic alterations that lead to resistance to treatment with CDK4/6 inhibitors (*Rubin et al., 2020*; *Cappell et al., 2018*; *Cappell et al., 2016*; *Zatulovskiy et al., 2020*). Thus, while G1 progression has been the focus of intense study, our understanding of its regulation remains incomplete.

The Polycomb Repressive Complex 2 (PRC2) was initially identified in *Drosophila* as a developmental regulator that represses the expression of Hox genes (*Lewis, 1978*). The PRC2 complex is conserved in throughout eukaryotes (*Dumesic et al., 2015*; *Grimaud et al., 2006*; *Jamieson et al., 2013*; *Schuettengruber et al., 2017*; *Kuroda et al., 2020*) and catalyzes the mono-, di-, and tri-methylation of Histone 3 Lysine 27 (referred to collectively as H3K27me3, the fully methylated form of H3K27) and thereby acts as a transcriptional repressor (*Schuettengruber et al., 2017*). The core PRC2 complex is composed of a H3K27me3 'reader' EED, a scaffold protein SUZ12, and the catalytic subunit EZH2 (or the more poorly expressed and less catalytically active paralog EZH1; *Son et al., 2013*). This core complex is capable of catalyzing H3K27me3 deposition and chromatin association, but how PRC2 achieves full spatiotemporal regulation of chromatin localization and catalytic activity has been an area of active investigation. Recent studies have identified additional accessory factors that modify the localization and enzymatic activity of these core components (*Hauri et al., 2016*). The associated auxiliary factors define different PRC2 subcomplexes, which are called PRC2.1 and PRC2.2, based on the composition of the subunits associated with the core PRC2 complex (reviewed in *van Mierlo et al., 2019*; *Piunti and Shilatifard, 2021*). In addition to the core PRC2 subunits, PRC2.1 consists of two modules, one module containing a Polycomb-like (PCL) protein PHF1, MTF2, or PHF19 and a second module of either PALI1/2 or EPOP. The more homogenous PRC2.2 always consists of the core PRC2 subunits in complex with both JARID2 and AEBP2. The role of these complexes in different cellular processes and contexts is debated. Despite the lack of an a clear analogous sequence to the Polycomb Response Elements which promotes PRC2 chromatin association in *Drosophila* (*Brown et al., 2018*; *De et al., 2020*), the presence of a DNA-binding extended homology domain in each PCL protein has been proposed to recruit PRC2.1 to unmethylated CpG islands and establish H3K27me3 (*Perino et al., 2018* ; *Li et al., 2017*). In contrast, PRC2.2 localizes to sites utilizing pre-existing mono-ubiquitinated H2AK119 (H2AK119ub1), which is deposited by the PRC1 complex (*Barbour et al., 2020*; *Cooper et al., 2016*; *Kalb et al., 2014*; *Glancy et al., 2023*), through a ubiquitin interaction motif contained within JARID2 (*Cooper et al., 2016*; *Kalb et al., 2014*; *Chen et al., 2018*; *Kasinath et al., 2021*). Regardless of their specific roles in the propagation of H3K27me3 histone marks,

members of both PRC2.1 and PRC2.2 have been implicated as both positive and negative regulators of stem cell maintenance, differentiation, and cancer, depending on the cellular context (*Piunti and Shilatifard, 2021*; *Su et al., 2015*; *Shirato et al., 2009*; *Adhikari and Davie, 2018*; *Petracovici and Bonasio, 2021*; *Loh et al., 2021*; *Rothberg et al., 2018*; *Ngubo et al., 2023*). All of the PRC2 core subunits (EZH2, SUZ12, and EED) have been shown to inhibit that transcription of both positive and negative regulators of G1/S progression, including the CDK4/6 protein inhibitor p16 (*Bracken et al., 2007*; *Bracken et al., 2006*; *Bracken et al., 2003*; *Adhikari et al., 2019*; *Adhikari and Davie, 2020*). However, the net result of these opposing effects on cell cycle progression, and the contribution of the individual subcomplexes to this regulation, remains unclear.

To identify novel regulators of cellular proliferation, we utilized a whole-genome chemogenetic approach to identify genes that sensitized or lent resistance to inhibitors of different cell cycle stages. We uncovered novel resistance mechanisms to three known inhibitors of cell cycle progression in the human haploid cell line HAP1. This approach revealed that mutations in mitochondrial function or the Polycomb complexes rescued slow proliferation in palbociclib. We could recapitulate these positive genetic interactions pharmacologically using small molecule inhibitors of either PRC2 activity or mitochondrial respiration. Loss of core PRC2 members or PCL subunits of the PRC2.1 subcomplex, particularly MTF2, resulted in resistance to palbociclib, while loss of PRC2.2-specific subunits resulted in sensitivity. Data from CUT&RUN and RNA sequencing experiments performed on clonal MTF2Δ and JARID2Δ knockout mutant cell lines suggest that PRC2.1 plays a more critical role in repressing gene expression when compared with PRC2.2 in HAP1 cells, particularly at promoters containing CpG islands. D-type cyclins are among the genes that are repressed by PRC2.1 and loss of MTF2 results in increased expression of both CCND1 and CCND2 through loss of H3K27me3 in their promoters. This increased expression resulted in an apparent increase in CDK4/6 kinase activity and S-phase entry of cells, driving resistance to CDK4/6 inhibition. Our results suggest that MTF2-containing PRC2.1 plays a strong role in G1 progression in a number of cellular contexts.

## Results
### Chemogenetic CRISPR–Cas9 screen utilizing cell cycle inhibitors identified novel players in the cell cycle

Recently, CRISPR–Cas9 knockout genetic screens have emerged as a powerful way in which to probe genetic interactions (*Horlbeck et al., 2018*; *Przybyla and Gilbert, 2022*; *Bock et al., 2022*), with the haploid human cell line HAP1 serving a popular model for these studies (*Hundley et al., 2021*; *Stok et al., 2023*; *Aregger et al., 2020*; *Hundley and Toczyski, 2021*; *Olbrich et al., 2019*; *Llargués-Sistac et al., 2023*). To identify novel genes involved in cell cycle regulation, we carried out genome-wide CRISPR–Cas9 chemogenomic screens in HAP1 cells treated with each of three well-characterized inhibitors of cell cycle progression: palbociclib (a CDK4/6 and G1 progression inhibitor), colchicine (a microtubule polymerization and mitosis inhibitor), and camptothecin (a Topoisomerase I and S/G2 inhibitor). We used a concentration for each inhibitor that reduced cellular proliferation by 30–50% (*Figure 1—figure supplement 1A*; see also *Hundley et al., 2021*) and confirmed their effects on cell cycle progression (*Figure 1—figure supplement 1B*). We then performed a CRISPR–Cas9 whole-genome screen for each of the three inhibitors (*Figure 1A*) by introducing the Toronto Knockout Library (*Hart et al., 2017*) via lentiviral transduction into an HAP1 cell line constitutively expressing Cas9. Following puromycin selection for 2 days, cells were propagated in either DMSO (Mock) or in the presence of drug (Treated) for 18 days. Following propagation, genomic DNA was extracted from the initial and final pools and subjected to deep sequencing, and gene–compound interactions were determined using the Orobas pipeline (Source Code File 1, *Supplementary file 1*, *Supplementary file 2*). A gene was considered as being significantly enriched or de-enriched in a given condition if both the Loess-adjusted differential gene effect between mean Treated and Mock control was ±0.5 and the and a false discovery rate (FDR) <0.4.

This approach resulted in the recovery of predicted compound–gene interactions demonstrating the robustness of both the screen and our analysis approach. For example, targeting of genes known to play roles in DNA damage repair (DDR) (*Groelly et al., 2023*; *Su et al., 2020*), including RAD54L, MUS81, and 16 proteins in the Fanconi Anemia pathway, strongly sensitized cells to camptothecin, which generates protein–DNA adducts (*Figure 1B, C*). The molecular target of camptothecin, TOP1

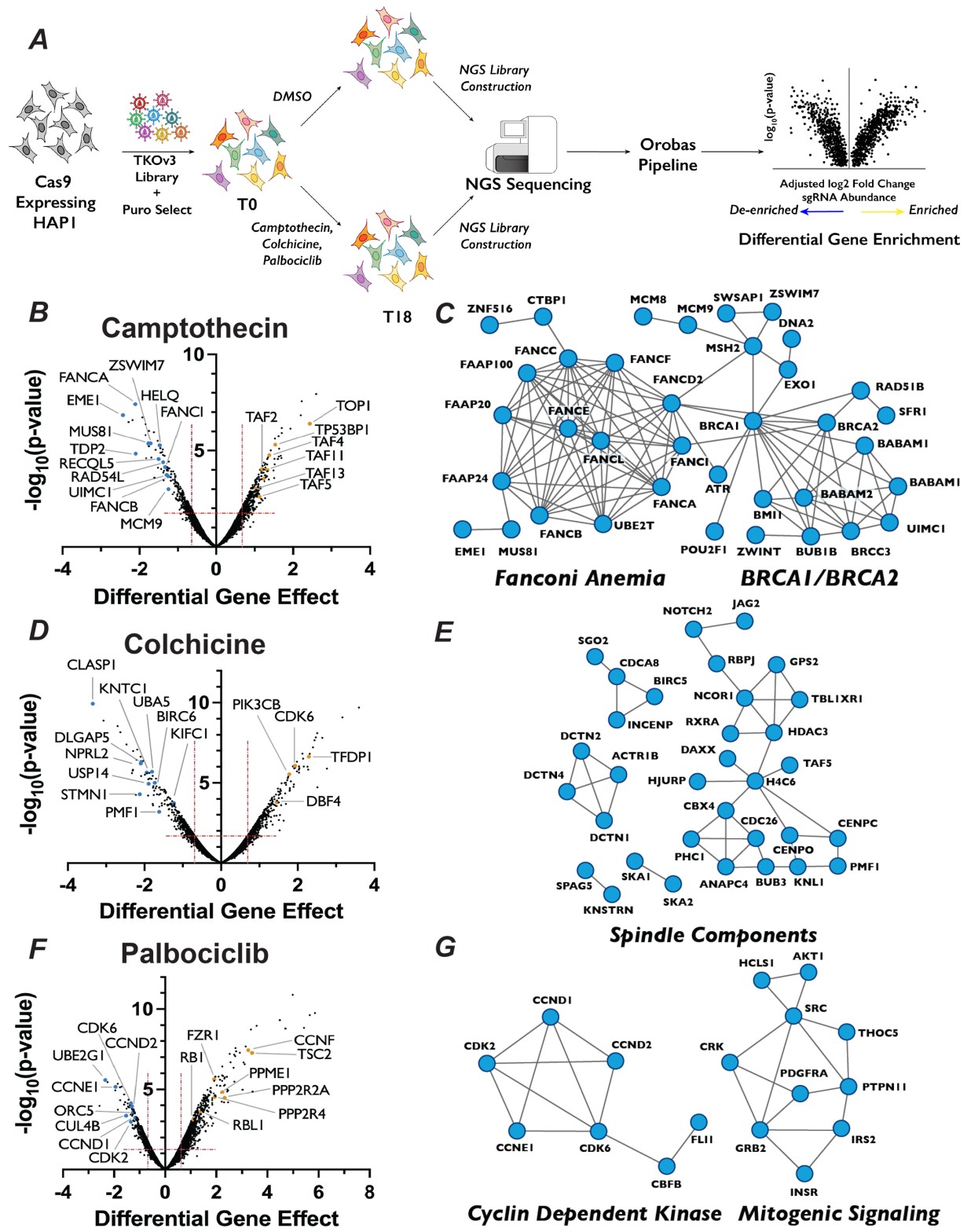

**Figure 1.** Chemogenetic CRISPR–Cas9 screen to study cell cycle progression. (**A**) Schematic of chemogenetic CRISPR–Cas9 screen. (**B**) Volcano plots of camptothecin chemogenetic screen results. The 'Differential Gene Effect' was plotted against the $-\log_{10}$(p-value) for this effect for each gene targeted in the screen, as calculated by the Orobas pipeline. Red dotted line indicates the established cut-off. Highlighted dots are genes with known roles in response to each treatment, with blue or yellow dots indicate genes that when inactivated resulted in sensitivity or resistance, respectively, to camptothecin. (**C**) Representative STRING analysis networks for protein complexes with known roles in pathways that we identified as sensitive in the

*Figure 1 continued on next page*

*Figure 1 continued*

camptothecin chemogenetic screen. Blue dots in the STRING network indicate genes that when inactivated resulted in sensitivity to camptothecin.
(D) Same as in (B) but for colchicine chemogenetic screen results. (E) Same as in (C) but for colchicine screen results. (F) Same as in (B) but for palbociclib chemogenetic screen results. (G) Same as in (C) but for palbociclib screen results.

The online version of this article includes the following figure supplement(s) for figure 1:

**Figure supplement 1.** Dosing to determine inhibitor concentration for chemogenetic screens.

(*Hsiang et al., 1985*), is the most resistant gene in the camptothecin screen, as are proteins involved in p53 transcriptional regulation, such as TP53BP1 and STAGA members TAF2, TAF4, TAF5, TAF11, and TAF13 (*Figure 1B*). Similarly, colchicine sensitized cells to the mutation of genes encoding proteins involved in mitotic spindle assembly, nuclear division and cytoskeletal assembly, such as CLASP1, DLGAP5, and KNTC1 (*Figure 1D, E*). Interestingly, inactivation of genes involved in the adaptive immune system, such as BIRC6, UBA5, and USP14, also resulted in sensitivity to colchicine. This observation is intriguing, as colchicine is used clinically as an immunomodulator in the treatment of gout (*Dalbeth et al., 2014*). CCNE1, CDK6, CDK2, CCND2, and CCND1, all of which are integral to promoting the G1/S phase transition, ranked as the 2nd, 24th, 27th, 29th, and 46th most important genes for palbociclib sensitivity, respectively (*Figure 1F, G*). CCND1 and CCND2 bind either CDK4 or CDK6, the molecular targets of palbociclib, whereas CDK2 and CCNE1 form a related CDK kinase that promotes the G1/S transition. Similarly, cells with sgRNAs targeting RB1, whose phosphorylation by CDK4/6 is a critical step in G1 progression, displayed substantial resistance to palbociclib. The recovery of genes known to function in the relevant biological processes supports the strength of this dataset and bolstered our confidence to use the results obtained to identify novel chemical–genetic interactions.

## Chemogenetic screen uncovered novel genetic interactions involved in response to inhibitors of cell cycle progression

To identify genes whose inactivation rendered cells sensitive or resistant to a specific cell cycle perturbation, we compared how the Orobas-calculated differential gene effect for a given targeted gene varied in each compound across our CRISPR–Cas9 screen. The majority of genes that conferred either sensitivity or resistance were specific to only one cell cycle inhibitor, with little overlap between the multiple conditions, suggesting that we identified genes that play roles in distinct biological processes (*Figure 2A*, *Figure 2—figure supplement 1*). For example, genes encoding DNA repair proteins, mitotic spindle components, and CDK2/4/6 holoenzyme components were only required for proliferation in camptothecin, colchicine, and palbociclib, respectively. We found that only 13 and 20 genes resulted in sensitivity or resistance, respectively, in all the compounds tested and were deemed nonspecific to a given condition, and excluded from any further analysis (see *Supplementary file 2*).

We next turned our attention to unexpected and novel compound–gene interactions. To probe these interactions, we analyzed genes that significantly altered response to our three compounds using the gene annotation and analysis portal Metascape and the protein–protein interaction network analysis STRING. In addition to DDR genes, Metascape and STRING analysis of the results of our camptothecin treatment revealed de-enrichment for sgRNAs targeting genes encoding members of the KICSTOR complex (KPTN, SZT2, ITFG2, and KICS2), which negatively regulates the mTOR pathway. In contrast, sgRNAs targeting of genes involved in RNA metabolism and chromatin organization increased resistance to this drug (*Figure 2B–D*). It has been suggested that mTOR is involved in attenuating the DDR response through phosphorylation of RNF168, leading to its degradation (*Xie et al., 2018*), which could provide one explanation of the observed sensitivity. The loss of genes involved in chromatin structure and the metabolism of RNA conferred resistance to camptothecin is unexpected, given that both these processes have been implicated in DNA repair after damage (*Bader et al., 2020*; *Stadler and Richly, 2017*). Genes whose inactivation enhanced sensitivity to colchicine included those involved in the amino acid starvation response (DEPDC5 TSC1, SZT2, and NPRL2) and mRNA splicing (SNRPB2, SF3B2, PPIL1, RBM22, and DHX35), while mutation of genes that control vesicle trafficking (VPS16, VPS18, VPS29, VPS41, VPS51, and VPS52) or encode members of the Mediator complex (CCNC, CDK8, and MED26, MED1, MED7, MED12, MED18, and MED11) attenuated the cellular response to the drug. Unexpectedly, inactivation of genes encoding members

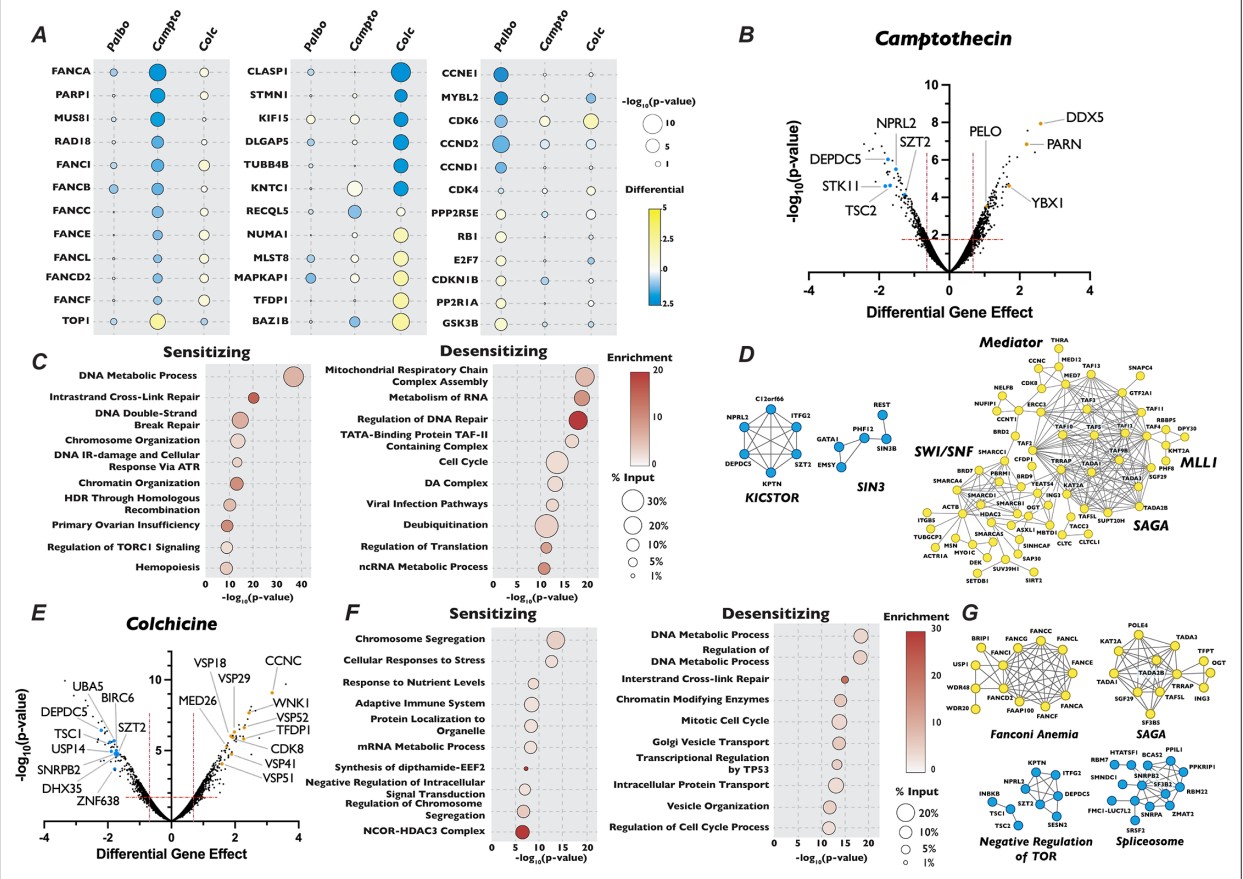

**Figure 2.** Analysis of camptothecin and colchicine chemogenetic screen reveals novel players in cell cycle regulation. (**A**) Dot plot comparison of the effect of gene mutation across three different screen conditions. Circle color indicates the strength of the positive or negative differential gene effect, circle size indicates the $-\log_{10}$(p-value) of the sgRNA enrichment. (**B**) Volcano plot of genes identified in the camptothecin chemogenetic screen, plotted as in *Figure 1B* with highlighted dots representing novel genes identified in the camptothecin screen. (**C**) Dot plot of Metascape analysis of significant genes that sensitized or de-sensitized cells to camptothecin. The $-\log_{10}$(p-value) of each term was plotted the enrichment was indicated by color of circle and the percentage of the input of genes associated with a given term is indicated by the size of the circle. (**D**) STRING analysis of genes identified from the analysis of the camptothecin screen. (**E**), (**F**), and (**G**) Same as in (**B**), (**C**), and (**D**) except for the colchicine screen.

The online version of this article includes the following figure supplement(s) for figure 2:

**Figure supplement 1.** Unique and shared genes identified as significantly enriched or de-enriched in chemogenetic screens.

of the TP53 signaling pathway (TFDP1 and HIPK2), SAGA H3 acetylation complex components (KAT2A, TRRAP, TADA3, TAF5L, TADA2B, SGF29, and TADA1), and the Fanconi Anemia complex (FANCA, FANCC, FANCE, FANCF, FANCG, FANCL, and FAAP100), all implicated in DDR, resulted in resistance to colchicine (*Figure 2E–G*). Sensitivity to palbociclib was enhanced in cells expressing sgRNAs targeting H4 acetylation, positive regulators of Pol II transcription and regulators of DDR (*Figure 3A, B*), although this sensitivity was much weaker than that seen with DNA damaging agents. This observation is consistent with long-term treatment with palbociclib inducing DNA damage, as has been suggested by a number of recent publications (*Crozier et al., 2023*; *Crozier et al., 2022*; *Wilson et al., 2023*). Unexpectedly, Metascape analysis of our palbociclib chemogenetic screen revealed that sensitivity to palbociclib was enhanced when genes involved in chromatin organization were targeted (*Figure 3A*). Inactivation of members of the SIN3 histone deacetylase (SIN3B, SINHCAF, and ARID4B), the NuA4 histone acetyltransferase (ING3, DMAP1, MORF4L2, YEATS4, and VPS73), the STAGA histone acetyltransferase (KAT2A, TADA1, TADA2B, TAF5L, and SUPT20H), and the Mediator (MED13, MED25, MED10, MED15, TAF7, TAG13, and CCNC) complexes all resulted in palbociclib sensitivity (*Figure 3B, C*). Each of these protein complexes promotes gene expression, suggesting that palbociclib sensitivity might be a result of a reduction in the transcription of genes important for the G1/S transition.

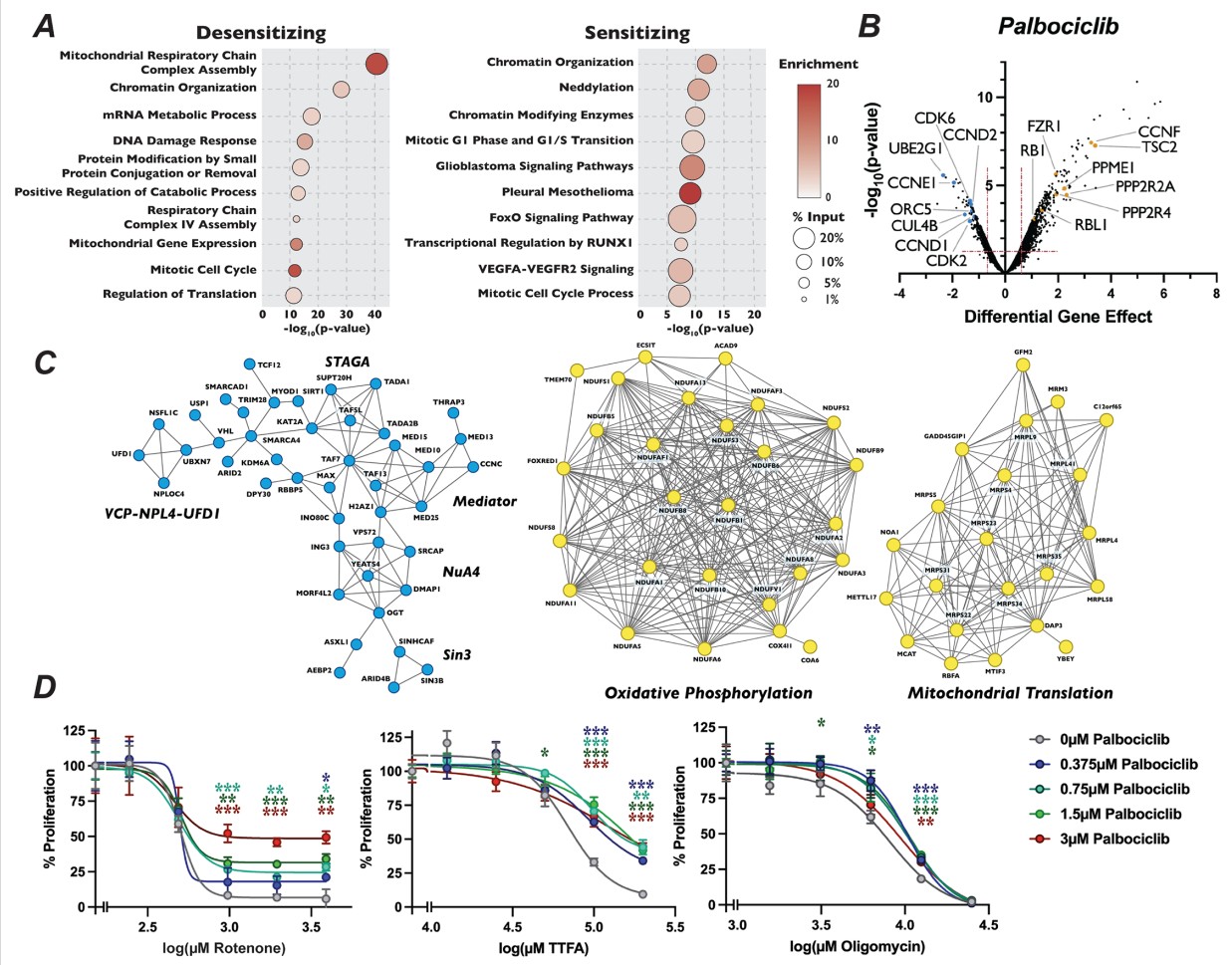

**Figure 3.** Mutation of mitochondria genes attenuates the sensitivity to palbociclib. (**A**) Dot plot of the −log₁₀(p-value) Metascape analysis of significant genes in the palbociclib chemogenetic screen. The enrichment of a given term is indicated by color of circle and the percentage of the input is indicated by the size of the circle. (**B**) Volcano plot of genes identified from our analysis of the palbociclib screen, plotted as in *Figure 1D*, with highlighted dots representing novel genes. (**C**) STRING networks of novel protein complexes identified in palbociclib screen. Dots in the STRING network indicate genes that when inactivated resulted in sensitivity (blue) or resistance (yellow) to palbociclib. (**D**) Dose–response curve of palbociclib-induced proliferation rescue in combination with oxidative phosphorylation inhibitors by PrestoBlue assay. Cells were grown in palbociclib with or without increasing concentrations of rotenone, thenoyltrifluoroacetone (TTFA), or oligomycin. Data represent mean of three technical replicates, normalized to the initial dose of each inhibitor in indicated concentration of palbociclib, ±StdDev. *p-value <0.05, **p-value <0.005, ***p-value <0.0005, n.s.: not significant, two-tailed unpaired Student's *t*-test.

The online version of this article includes the following figure supplement(s) for figure 3:

**Figure supplement 1.** Alternative representation of dose–response curves for combination of oxidative phosphorylation and palbociclib.

Because mechanisms of clinical resistance to palbociclib are an area of active investigation, we turned our attention to focus on these pathways. Metascape analysis of genes whose loss conferred palbociclib resistance was highly enriched for splicing factors, oxidative phosphorylation and mitochondrial translation, in addition to chromatin modification (*Figure 3A*). STRING analysis of the high-confidence, physical interactions of proteins important for palbociclib sensitivity revealed multiple highly connected interaction networks (*Figure 3C*). Strikingly, almost 25% (170 out of the 689) of the genes whose mutation conferred unique resistance to palbociclib have terms associated with mitochondrial respiratory chain complex assembly, ATP synthesis, or mitochondrial gene expression. Specifically, we see many components implicated in assembly of Mitochondrial Respiratory Chain Complex I and IV, as well as core mitochondrial ribosome and mitochondrial translation initiation and termination (*Figure 3A–C*). To confirm this positive genetic interaction between mitochondrial homeostasis and resistance to palbociclib, and to dissect whether specific electron transport chain steps might

be implicated in this resistance, we asked whether chemical inhibition of oxidative phosphorylation could rescue sensitivity to palbociclib. To target different stages of the oxidative phosphorylation, we employed rotenone, TTFA, and oligomycin, which inhibit Complex I, Complex II, and ATP synthase, respectively. Cells were grown in the presence of palbociclib alone or in combination with each drug for 48 hr and viability was determined by PrestoBlue assay. Cells exposed to rotenone, TTFA, and oligomycin displayed dose-dependent, positive proliferation interactions when treated in combination with palbociclib (*Figure 3D*, *Figure 3—figure supplement 1A–C*), indicating that combined inhibition of oxidative phosphorylation and CDK4/6 activity mutually rescue the proliferation defect imposed by agents targeting the other process. While alternative explanations could explain the observed novel chemical–genetic interactions we uncovered here, such as either changes in phenotypic lag rates due to alterations in protein stability or more general screen variability (*Rahman et al., 2021*), these results suggest a connection between mitochondrial gene function and CDK4/6 inhibitors.

## Polycomb repressive complex components display differing responses to palbociclib treatment

Intriguingly, inactivation of EZH2, SUZ12, and EED, the three core members of the PRC2 complex, resulted in profound resistance to palbociclib, being the eighth, fourth, and third strongest resistance hits out of the 18,053 genes examined when ranked by the score of differential effect (*Figure 4A*, *Supplementary file 1*). Mutation of RBBP7, which associates with the core PRC2 complex (*Schuettengruber et al., 2017*) along with a number of histone deacetylases (*De La Fuente, 2014*), also de-sensitized cells to palbociclib, but to a more modest extent. Satisfyingly, inactivation of RING1, RNF2, and PCGF6, which are members of PRC1, also displayed resistance to palbociclib. The PRC1 complex contains a ubiquitin ligase that works in concert with PRC2 through H2AK119ub1 deposition, a histone mark that influences both PRC2 chromatin localization and catalytic activity (*Barbour et al., 2020*; *Kalb et al., 2014*). As expected, PRC1 and PRC2 components identified in our palbociclib chemogenetic screen formed a highly interconnected STRING physical interaction network (*Figure 4B*), indicating that loss of either H3K27me3 or H2AK119ub1 reduced sensitivity to this drug. In contrast, inactivation of genes encoding OGT, ASXL1, and HAT1, which are members of the H2AK119ub1 deubiquitinase complex that opposes PRC2-mediated gene repression (*Chittock et al., 2017*), resulted in sensitivity to palbociclib (*Figure 4A*). Importantly, no component of any PRC1 or PRC2 core complex displayed significant resistance or sensitivity to camptothecin and colchicine in our chemogenetic screens (*Figure 4C*), implicating PRC2 in the regulation of G1 specifically, and not to other phases of the cell cycle or the DNA damage response pathway. PR-DUB components ASXL1 and OGT did show resistance to camptothecin, consistent with their role in repressing the homologous recombination DNA repair pathway (*Ismail et al., 2014*). We sought to confirm the role of the core PRC2 complex in palbociclib resistance by treating cells with combinations of palbcociclib and the EZH2 inhibitor GSK126 using a quantitative Crystal Violet assay. After 9 days of drug combination treatments, we found that cells treated with increasing doses of GSK126 withstood palbociclib-induced proliferation suppression (*Figure 4D*), though even the highest dose utilized of 5 µM GSK126 had some synergistic effects with low doses of palbociclib. These results confirmed that inactivation of the PRC2 core complex, either through genetic inactivation or chemical inhibition, resulted in resistance to palbociclib.

The PRC2 core binds to auxiliary proteins to create biochemically distinct subcomplexes, termed PRC2.1 and PRC2.2 (*Hauri et al., 2016*; *van Mierlo et al., 2019*). These alternative complexes are thought to modify the chromatin localization and enzymatic activity of PRC2, reenforcing existing H3K27me3 in certain contexts (*Healy et al., 2019*; *Zhu et al., 2022a*; *Youmans et al., 2021*), while initiating H3K27me3 deposition at new loci in others (*Glancy et al., 2023*; *Petracovici and Bonasio, 2021*). Mutation of the PRC2.1 complex members PHF1, MTF2, PHF19, and EPOP/C17orf96 all display significant resistance to palbociclib, with MTF2 being the strongest of these (*Figure 4A, C*). Conversely, targeting the genes encoding the PRC2.2-specific accessory proteins AEBP2 or JARID2 resulted in enhanced or neutral palbociclib sensitivity, respectively. These data suggest that PRC2.1 plays a previously uncharacterized role in promoting G1 progression, while PRC2.2 antagonizes it. To confirm the results from our palbociclib chemogenetic screen, we generated polyclonal knockout mutant pools of the individual core and accessory proteins of PRC2. We generated these populations by independently infecting three distinct sgRNAs targeting genes for each PRC2 complex

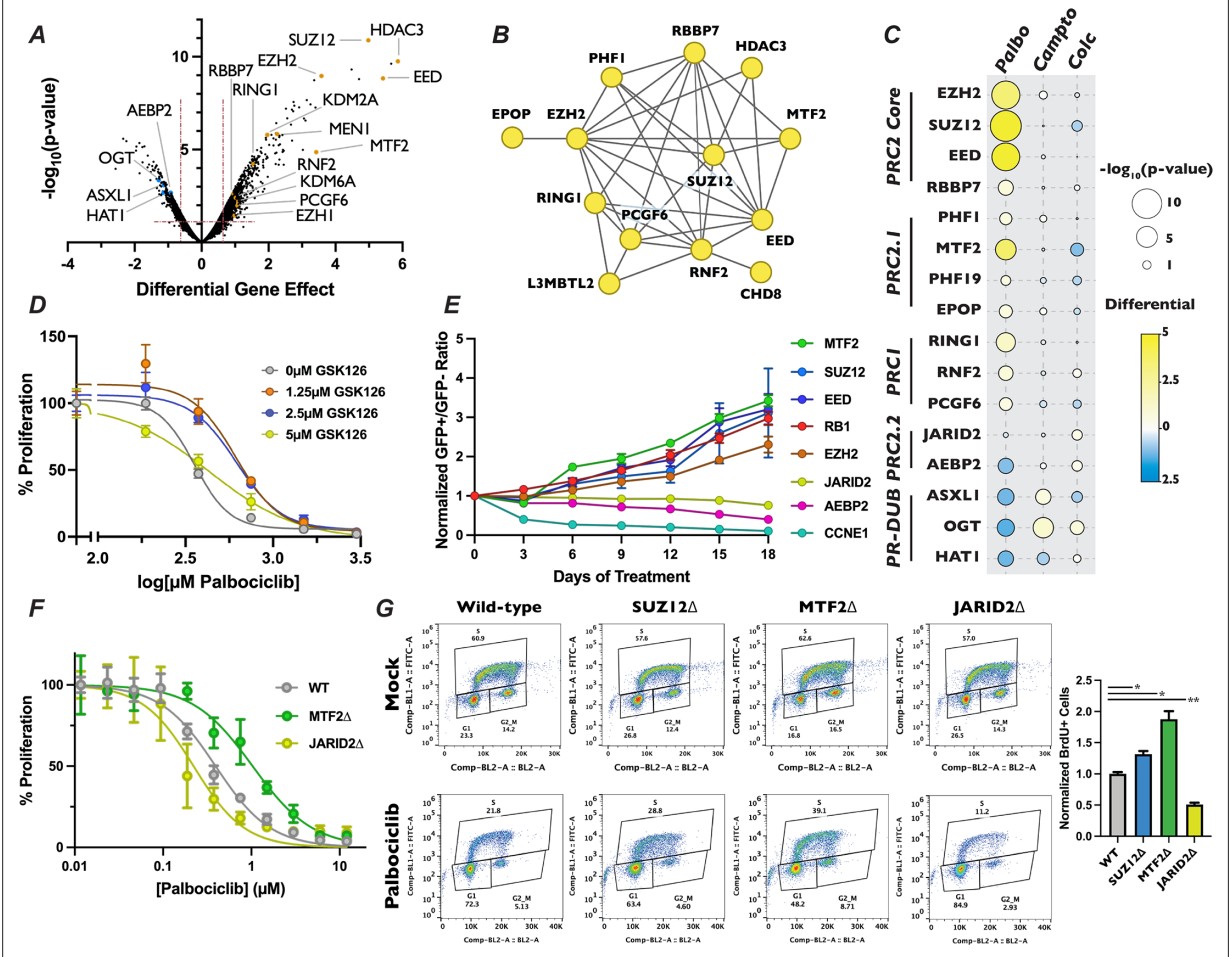

**Figure 4.** Loss of polycomb repressive complex components display specific resistance to palbociclib. (**A**) Volcano plot as in *Figure 3B* except with members of PR-DUB, PRC1, and PRC2 highlighted. (**B**) STRING analysis network of PRC components. Yellow dots indicate that inactivation of these genes conferred resistance to palbociclib. (**C**) Dot plot of comparison of the effect of PRC2 complex member gene mutation across three different screen conditions, as in *Figure 2B*. (**D**) Dose–response curve of palbociclib-induced proliferation inhibition rescue with GSK126 by Crystal Violet assay. Data were normalized to untreated cells and represents the mean of three technical replicates, ±StdDev. (**E**) Results of competitive proliferation assay for each indicated time point, normalized to the initial GFP⁺/GFP⁻ ratio of the pool. The performance of each sgRNA in 1.5 µM palbociclib versus Mock is shown, after normalizing to control sgRNAs, ± d Graduate Program, University of CalifSEM of the GFP⁺/GFP⁻ ratios of three independent sgRNAs. (**F**) Dose–response curve of palbociclib-induced proliferation inhibition in MTF2Δ and JARID2Δ cells by Crystal Violet assay. Data represents mean staining of three monoclonal knockout cell lines, ±StdDev. (**G**) BrdU incorporation assay for wild-type, SUZ12Δ, MTF2Δ, and JARID2Δ cell lines. *Left* – Representative BrdU incorporation versus propidium iodide flow cytometry traces. *Right* – Quantification of BrdU incorporation assay, mean of S-phase cells in three knockout lines ±StDev. *p-value <0.05, **p-value <0.005, n.s.: not significant, two-tailed unpaired Student's *t*-test.

The online version of this article includes the following source data and figure supplement(s) for figure 4:

**Figure supplement 1.** Competitive proliferation assay to determine resistance of PRC2 component mutants to CDK4/6 inhibitors.

**Figure supplement 1—source data 1.** Original files for western blots shown in *Figure 4—figure supplement 1B*.

**Figure supplement 1—source data 2.** Original files for western blots shown in *Figure 4—figure supplement 1B*, indicating relevant band.

**Figure supplement 2.** MTF2Δ mutant cell lines display resistance CDK4/6 inhibitors in competitive proliferation assay.

**Figure supplement 3.** PARP cleavage does not change in MTF2Δ or JARID2Δ knockout mutant cell lines upon palbociclib treatment.

**Figure supplement 3—source data 1.** Original files for western blots shown in *Figure 4—figure supplement 3*.

**Figure supplement 3—source data 2.** Original files for western blots shown in *Figure 4—figure supplement 3*, indicating relevant band.

member, or positive and negative control genes, in GFP-positive and doxycycline-inducible Cas9 cells and induced DNA cleavage for 3 days (henceforth referred to as pooled knockouts). We performed western blots to confirm reduction in protein levels for the genes targeted by the sgRNAs used to generate our pooled knockouts (*Figure 4—figure supplement 1B*). After confirming reduction in

the targeted proteins, we carried out a competitive proliferation assay using these pooled knockouts (schematic in *Figure 4—figure supplement 1A* or see *Hundley et al., 2021*). Briefly, GFP-positive pooled knockouts were mixed with GFP-negative wild-type cells and propagated in the presence or absence of palbociclib. The ratio of GFP-positive to GFP-negative cells was recorded every 3 days by flow cytometry for 18 days. Pools containing sgRNAs against EZH2, SUZ12, EED, and MTF2 all displayed resistance to palbociclib, similar to the level observed with sgRNAs targeting RB1, our positive control for palbociclib-induced proliferation defects (*Figure 4E*). Conversely, compared with the dramatic reduction seen in pools transduced with CCNE1 sgRNAs, our positive control for enhanced palbociclib sensitivity CCNE1, sgRNAs targeting PRC2.2 components showed a slight, but statistically signification reduction in proliferation in palbociclib over the 18-day assay (AEBP2: p-value = 0.002 and JARID2: p-value = 0.0148, unpaired two-tailed Student's *t*-test). Thus, we confirmed the results of our chemogenetic screen that MTF2-containing PRC2.1 inhibits G1 progression, while PRC2.2 could promote it.

To further interrogate the role of PRC2.1 and PRC2.2 in the regulation of G1 progression, we generated SUZ12, MTF2, and JARID2-null monoclonal cell lines (which we will refer to as SUZ12Δ, MTF2Δ, and JARID2Δ). We chose to mutate MTF2 to probe the function of PRC2.1, as it has been shown to be the most highly expressed and predominant PCL subunit associated with the PRC2 core complex in a variety of contexts (*Oliviero et al., 2016*; *Smits et al., 2013*). Furthermore, we selected SUZ12 for inactivation out of the core PRC2 complex members, and not the catalytic subunit EZH2, because the presence of the EZH2 paralog EZH1 might compensate for loss of EZH2 (*Jadhav et al., 2020*). Additionally, SUZ12 has a critical role in bridging accessory proteins with the catalytic core in all known PRC2 complexes (*Petracovici and Bonasio, 2021*). MTF2Δ cells displayed resistance to palbociclib when compared with wild-type cells in a 9-day quantitative Crystal Violet assay (MTF2Δ $IC_{50}$ = 1.033 μM, wild-type $IC_{50}$ = 0.3936 μM) while JARID2Δ cells were slightly more sensitive than wild-type (JARID2Δ $IC_{50}$ = 0.2216 μM) (*Figure 4F*). In addition to showing sensitivity to palbociclib, MTF2Δ cells also displayed resistance to, ribociclib and abemaciclib, two CDK4/6 inhibitors that are structurally related to palbociclib, in a competitive proliferation assay (*Figure 4—figure supplement 2*). These results confirmed our screen results that mutation of MTF2 leads in CDK4/6 inhibitor resistance with verified clonal mutants.

Palbociclib exerts its chemotherapeutic effects by inducing a G1 arrest and senescence in tumor cells with a functional RB-E2F pathway (*Zhu et al., 2022b*; *Schoninger and Blain, 2020*; *Dean et al., 2010*). However, a recent report demonstrates that palbociclib treatment induces both G1 arrest and apoptosis through the increase in DNA damage in cultured cells (*Wang et al., 2021*), introducing the possibility that PRC2.1 could be altering regulators of the DDR pathway, resulting in the observed palbociclib resistance. To determine if inactivation of PRC2.1 or PRC2.2 altered cell cycle progression, we examined how wild-type, SUZ12Δ, MTF2Δ, and JARID2Δ cells responded to palbociclib-induced G1 arrest. To assess this directly, we performed a BrdU incorporation assay by growing each mutant for 24 hr in palbociclib, pulsed the cells with BrdU for 1 hr prior to harvest and then measured BrdU incorporation by flow cytometry. Each of the four cell lines had similar levels of BrdU incorporation in the absence of drug (*Figure 4G*). However, MTF2Δ and SUZ12Δ mutants displayed significantly more cells entering S-phase in the presence of palbociclib, while palbociclib treatment resulted in significantly fewer JARID2Δ mutants cells in S phase (*Figure 4G*). To rule out the possibility that cellular viability was not compromised in our monoclonal knockout cell lines, we used western blotting to monitor changes in PARP cleavage or increased BCL2L11/BIM expression, which both serve as apoptosis indicators (*Sionov et al., 2015*). There was no detectable basal increase in markers of apoptosis in the monoclonal knockout mutant cell lines or when cells were treated with palbociclib for 48 hr (*Figure 4—figure supplement 3*), supporting the conclusion that the resistance to palbociclib observed in the MTF2Δ and SUZ12Δ cells was due to the repressive and promoting role the MTF2-containing PRC2.1 and PRC2.2 complexes play, respectively, in the canonical CDK4/6-RB1-E2F pathway.

## PRC2.1 and PRC2.2 mutants display altered H3K27me3 and transcriptional landscapes

To determine why the mutation of PRC2 subcomplex components altered the cellular response to palbociclib, we sought to see how H3K27me3 levels and gene expression changed in MTF2Δ and

JARID2Δ cells. western blotting of total H3K27me3 levels in three independently generated clones indicated that there was no change in the bulk levels of H3K27me3 (*Figure 5A*), suggesting that any change of phenotype observed in the mutants was due to a change in the localization of this mark and not due to an overall reduction in its abundance. This is in contrast to SUZ12Δ cells, which displayed a significant reduction in the H3K27me3 mark (unpaired Student's *t*-test, p-value = 0.0104). To probe the changes in transcription and H3K27me3 distribution genome-wide, we generated CUT&RUN libraries with an anti-H3K27me3 antibody and RNA-Seq libraries from total RNA isolated from our MTF2Δ and JARID2Δ cell lines, grown either in the presence or absence of palbociclib for 24 hr. Changes in H3K27me3 levels and mRNA expression were determined by comparing the enrichment of reads in the MTF2Δ and JARID2Δ libraries to the wild-type cell line (*Supplementary file 3* and *Supplementary file 4*). Because cancer cells have been known to adapt to palbociclib treatment partially through changes to histone marks, chromatin structure and gene expression (*Watt et al., 2021*; *Pancholi et al., 2020*; *Guarducci et al., 2018*; *Turner et al., 2019*), we also investigated how both transcript levels and H3K27me3 distribution responded to treatment with palbociclib in our clonal knockout cell lines. Primary component analysis (PCA) of our called, reproducible H3K27me3 peaks and transcript abundance from our CUT&RUN and RNA-Seq data, respectively, showed a high percentage of variance between each of the genotypes tested, along with good clustering of repeats of the same genotype and treatments (*Figure 5—figure supplement 1C*), suggesting a shift in the epigenetic and transcriptional landscapes when either MTF2 or JARID2 are absent. PCA analysis of our RNA-Seq experiment revealed substantial shifts in variance between palbociclib- and Mock-treated samples for each genotype (*Figure 5—figure supplement 1C*, bottom), suggesting that exposure to palbociclib resulted in changes in gene expression, consistent with previous reports (*Ferguson et al., 2023*; *Lanceta et al., 2021*). However, the PCA of our CUT&RUN experiment did not reveal large differences in H3K27me3 distribution between palbociclib-treated and untreated samples (*Figure 5—figure supplement 1C* – top). In line with this observation, when we analyzed the change in distribution of H3K27me3 peaks between palbociclib- and Mock-treated cells using DESeq2, we found no significant changes in the location of H3K27me3 reproducible peaks in the presence or absence of palbociclib (data not shown). This suggests that MTF2Δ mutants are not resistant to palbociclib because MTF2 is required for a transcriptional adaptation to the drug, but instead because MTF2 alters expression of genes important for G1/S progression, even in unperturbed cells.

Due to the known role of PRC2 in repressing gene expression, we next asked how H3K27me3 distribution changed in promoters of genes. We defined promoters as 4 kb upstream and 1 kb downstream of all annotated transcription start sites, and calculated the total number of reads within each of these regions. Our parameters led to ~61,000 genomic regions being designated as promoters. In addition to annotated protein-coding genes, this included the promoters of non-coding transcribable units such as rRNA, miRNAs, lncRNAs, and pseudogenes. We observed a greater number of promoters with significantly decreased H3K27me3 ($\log_2$ fold-change [LFC] ±1, adjusted p-value <0.1) in the MTF2Δ compared to JARID2Δ cell lines (5808 vs 1034 promoters, respectively). Of these, 5149 promoters displayed MTF2-dependent H3K27me3, 392 were JARID2-dependent, and 629 were co-dependent on MTF2 and JARID2 for wild-type levels of H3K27me3 (*Figure 5B, D*). Consistent with the greater change in H3K27me3 signal at promoters in MTF2Δ cells, 733 versus 114 transcripts were significantly upregulated upon MTF2 versus JARID2 inactivation, respectively, with 666 transcripts that were exclusively MTF2-dependent, 47 transcripts that were exclusively JARID2-dependent, and 67 transcripts displaying co-dependence on both MTF2 and JARID2. These results indicate that the MTF2-containing PRC2.1 complexes affect the deposition of H3K27me3 in the promoter regions of more genes than the JARID2-containing PRC2.2, and therefore, are more important for transcriptional repression in HAP1 cells.

Given the diverse regulatory roles of PRC2 in different biological contexts, and the limited information on PRC2.1 and PRC2.2 outside of stem cells, we were curious to see what classes of genes were being differentially regulated in the MTF2Δ and JARID2Δ cell lines. Only ~30–40% of the promoters with significantly changed levels of H3K27me3 were upstream of protein-coding genes (*Figure 5—figure supplement 1A*), while ~80–90% of the significantly differentially expressed transcripts encoded proteins (*Figure 5—figure supplement 1B*). Therefore, we focused a Metascape analysis on the promoters and mRNAs of protein-coding genes with differential H3K27me3 and transcript levels, respectively (*Figure 5C*). Analysis of the promoters of genes with decreased H3K27me3

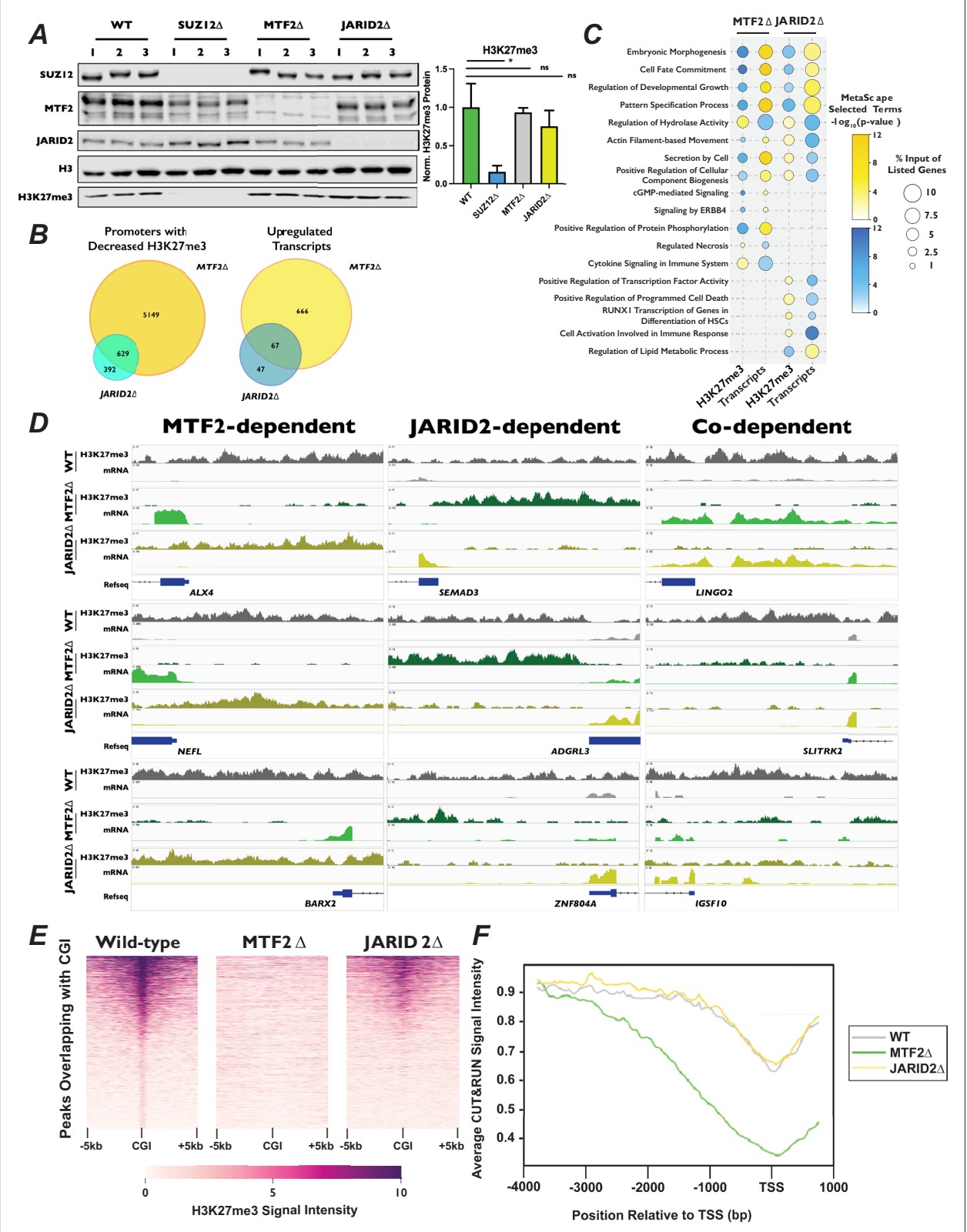

**Figure 5.** Polycomb 2.1 and PRC2.2 are differentially recruited to promoters with CpG islands. (**A**) *Left* – Western blots of wild-type, SUZ12Δ, MTF2Δ, and JARID2Δ cell extracts probed with the indicated antibodies. *Right* – Quantification of H3K27me3 signal intensity, normalized to H3, ±StDev. *p-value <0.05, n.s.: not significant, two-tailed unpaired Student's *t*-test. (**B**) Venn diagrams of MTF2Δ or JARID2Δ compared to wild-type cells of *left* – promoters with decreased H3K27me3 signal in CUT&RUN experiment or *right* – increased transcript levels in RNA-Seq. (**C**) Dot-plot of selected Metascape terms

*Figure 5 continued on next page*

*Figure 5 continued*

of protein-coding genes displaying significantly increased or decreased levels of H3K27me3 or transcripts. Color of the circle indicates the $-\log_{10}$(p-value) of the term and the size of circle indicates the percentage of the genes from the input list were represented in that term. (**D**) Genome browser traces of promoters with decreased H3K27me3 and increased mRNAs that were dependent on MTF2 (*left*), JARID2 (*center*), or on the presence either MTF2 or JARID2 (*right*). Tracks represent combined BED files from two clonal biological replicates. (**E**) Representative heatmap of H3K27me3 signal for 1877 peaks overlapping with CGI. Genomic regions are ordered by the H3K27me3 read density intensity in wild-type cells then plotted for the same loci in MTF2Δ and JARID2Δ cells. Plots are of one of two biological replicates. (**F**) H3K27me3 signal averaged for all CGI-containing promoters for wild-type, MTF2Δ, and JARID2Δ cells.

The online version of this article includes the following source data and figure supplement(s) for figure 5:

**Source data 1.** Original files for western blots shown in *Figure 5A*.

**Source data 2.** Original files for western blots shown in *Figure 5A*, indicating relevant band.

**Figure supplement 1.** Changes in H3K27me3 distribution and differentially expressed genes in MTF2Δ and JARID2Δ cells.

**Figure supplement 2.** Average H3K27me3 distribution over a 10 kb window for 1877 peaks overlapping with CGIs in MTF2Δ and JARID2Δ cells.

**Figure supplement 3.** Functional analysis of CGIs.

and increased transcript levels in both MTF2Δ and JARID2Δ lines were terms associated with embryonic morphogenesis, cell-fate commitment, and developmental growth, all processes previously been shown to be regulated, at least in part, by PRC2 (*German and Ellis, 2022*). Intriguingly, terms for genes that specifically displayed decreased promoters H3K27me3 and upregulated mRNA in MTF2Δ cells included the pro-growth pathways cGMP and ERBB4 signaling. Conversely, terms for genes which displayed increased promoter H3K27me3 signal and decreased transcript levels in JARID2Δ cells contained pathways that could reduce cellular proliferation and viability, such as positive transcriptional regulation of RUNX1 and positive regulators of program cell death. We also saw terms that had opposite effects on H3K27me3 and transcript levels in MTF2Δ compared to JARID2Δ cells, such as secretion by the cell and regulation of cellular component biogenesis, which could potentially exacerbate palbociclib-induced proliferation defects (*Uzhachenko et al., 2021*; *Franco et al., 2016*). Together, these data support the notion that MTF2 antagonizes cell proliferation in normal cellular conditions, while JARID2 promotes it.

## PRC2.1 and PRC2.2 mutants display differential H3K27me3 modification in promoters in cell cycle related genes with CpG islands

MTF2-containing PRC2.1 have been previously shown to localize to chromatin using a winged helix in its extended homology domain that has affinity for CG-rich sequences (*Perino et al., 2018*; *Li et al., 2017*), whereas PRC2.2 localization is dependent on chromatin context, specifically H2AK119ub1 deposited by PRC1 (*Cooper et al., 2016*; *Kalb et al., 2014*; *Glancy et al., 2023*). To determine whether CpG island targeting by PRC2.1 could help explain the palbociclib resistance we observed in the absence of MTF2, we identified and plotted 1877 peaks that overlapped with CpG islands in wild-type cells and had the greatest H3K27me3 signal in a 10-kb window surrounding the CpG islands. We then plotted the H3K27me3 signal observed in the MTF2Δ and JARID2Δ cells for these same loci. We observed a complete loss of H3K27me3 signal intensity at CpG islands in the MTF2Δ mutants, but only a partial loss at these loci in JARID2Δ cells (*Figure 5E*, *Figure 5—figure supplement 2*). When we expanded our findings genome-wide, we found a significant loss of H3K27me3 peaks at CpG islands in MTF2Δ cells (Fisher's exact test, odds ratio = 20.4, p-value $2.2 \times 10^{-308}$), compared with JARID2Δ, where this loss was much more modest (Fisher's exact test, odds ratio = 9.8, p-value = $6.5 \times 10^{-7}$). This result is consistent with the interpretation that the MTF2-containing PRC2.1 is required for all H3K27me3 deposition at CpG islands, whereas JARID2-containing PRC2.2 is only required to achieve full wild-type H3K27me3 levels at these sites.

CpG islands are a very common feature of mammalian promoters, with 50–70% human promoters estimated to contain at least one CpG island (*Zheng et al., 2021*). Since promoters are highly associated with CpG islands, we examined 2000 promoters with the highest level of H3K27me3 signal intensity that overlapped with CpG islands in wild-type cells, then plotted the H3K27me3 signal intensity at those same loci in our mutant cell lines (*Figure 5F*). Consistent with the result seen at CpG islands genome-wide, we observed a complete loss of high signal intensity in the MTF2Δ cells, but only a slight loss in JARID2Δ cells. When we averaged the H3K27me3 signal intensity over all 25,124

promoters that contain CpG islands, we observed a pattern of MTF2Δ cells having greatly decreased H3K27me3 levels in these regions, particularly surrounding the transcription start site while JARID2Δ cells were widely similar to wild-type cells (*Figure 5F*) in line with what was seen at CpG islands genome-wide. Reactome and MSigDB analysis of the promoters of protein-coding genes that over-lapped with CpG islands showed strong enrichment for terms associated with cell cycle and E2F target genes (*Figure 5—figure supplement 3A*) as well as enrichment binding E2F6 (*Figure 5—figure supplement 3B*), which both regulates transcription of G1 progression genes (*Giangrande et al., 2004*) and is a well-characterized component of Polycomb complexes (*Hauri et al., 2016*; *Shirahama and Yamamoto, 2020*). These results suggest that MTF2 is required for H3K27me3 deposition at promoters containing CpG islands involved in cell cycle regulation and can explain why MTF2Δ cells display a greater change in gene expression than do JARID2Δ cell lines.

## PRC2.1 represses expression of CCND1 and CCND2

Our CUT&RUN results suggest that MTF2-containing PRC2.1 impacts gene expression, at least in part, through deposition of H3K27me3 at promoters with CpG islands. Therefore, we hypothesized this PRC2 complex must be antagonizing G1 progression through repression of cell cycle-promoting genes. When inspecting the results of our CUT&RUN and RNA-Seq experiments, we found that the promoters of both CCND1 and CCND2 had lost H3K27me3 signal and displayed strong transcriptional induction in MTF2Δ cells (*Figure 6A*, *Figure 6—figure supplement 1A, B*), suggesting that the increase in these transcripts was due directly to a change in H3K27me3 in their promoters. In fact, while CCND1 and CCND2 were both among the most upregulated statistically significant transcripts within the MTF2Δ cell line, their transcription and promoter H3K27 methylation were unaltered in JARID2Δ cells (*Figure 6—figure supplement 1C–E*). Given that increased CCND1 levels are sufficient to drive increased CDK4/6 kinase activity, upregulation of these D-type cyclins is likely to be a significant contributor to the palbociclib resistance in MTF2Δ cells. DESeq2 analysis of H3K27me3 density in MTF2Δ cells displayed a statistically significant 4.3 and 2.7 $\log_2$ fold-decrease in H3K27me3 signal in the promoter region of CCND1 and CCND2, respectively, when compared to wild-type H3K27me3 levels (*Figure 6—figure supplement 1A*, *Supplementary file 3*), whereas changes in H3K27me3 levels in the CCND3 promoter were not statistically significant (*Supplementary file 3*). Given our observation that H3K27me3 signal is lost at CpG islands in MTF2Δ cells, we inspected the D-type cyclin promoters for CpG islands. Indeed, the regions upstream of all three D-type cyclins contained CpG islands, but CCND1 and CCND2 had regions of GC density about seven times larger (7460 and 6003 bp, respectively) than CCND3 (996 bp) (*Figure 6B*, *Figure 6—figure supplement 1F*). Furthermore, the promoter of CCND1 contained about twice as many CpG repeats than did CCND2 (575 vs 379) and about six times as many CpG repeats as CCND3 (575 vs 95). These results suggest that the levels of CCND1 and CCND2 mRNA transcripts, but not CCND3, were regulated by MTF2 in a CpG island-dependent manner.

We sought to confirm our observation that ablation of MTF2 resulted in increased levels of CCND1 and CCND2 protein. We generated pooled knockouts of MTF2, JARID2, and the core PRC2 components SUZ12, EZH2, and EED using three independent sgRNAs. In pooled knockouts of MTF2, EZH2, EED, and SUZ12, we observed an increase in both CCND1 and CCND2 protein levels by western blot, but not for CCND3 (*Figure 6C*, *Figure 6—figure supplement 2A*). Consistent with the results from our CUT&RUN and RNA-Seq datasets, we did not observe a significant change in either CCND1 or CCND2 levels in JARID2 pooled knockouts. We next examined mRNA and protein levels of the D-type cyclins in MTF2Δ and JARID2Δ clones by qRT-PCR and western blotting, respectively. Again clonal knockouts of SUZ12Δ and MTF2Δ, but not JARID2Δ lines, had increased mRNA (*Figure 6—figure supplement 2C*) and protein levels (*Figure 6—figure supplement 2B*) for both CCND1 and CCND2, but not CCND3. To determine whether other genes involved in the canonical CDK4/6-RB-E2F pathway were also altered, we examined mRNA and protein levels of known cell cycle regulators in our knockout cell lines. In contrast to CCND1 and CCND2, none of the E-type cyclins, CIP/KIP CDK inhibitors, RB1 or E2F proteins displayed significantly altered mRNA transcript abundance in our RNA-Seq experiment in either MTF2Δ or JARID2Δ lines (*Figure 6—figure supplement 3A*). To confirm that protein stability of these factors was not altered in our knockout lines, we also examined protein levels of a panel of known G1 regulators by western blot (*Figure 6—figure supplement 3B*). Similarly, we did not observe an increase in levels of any of the tested proteins, confirming that

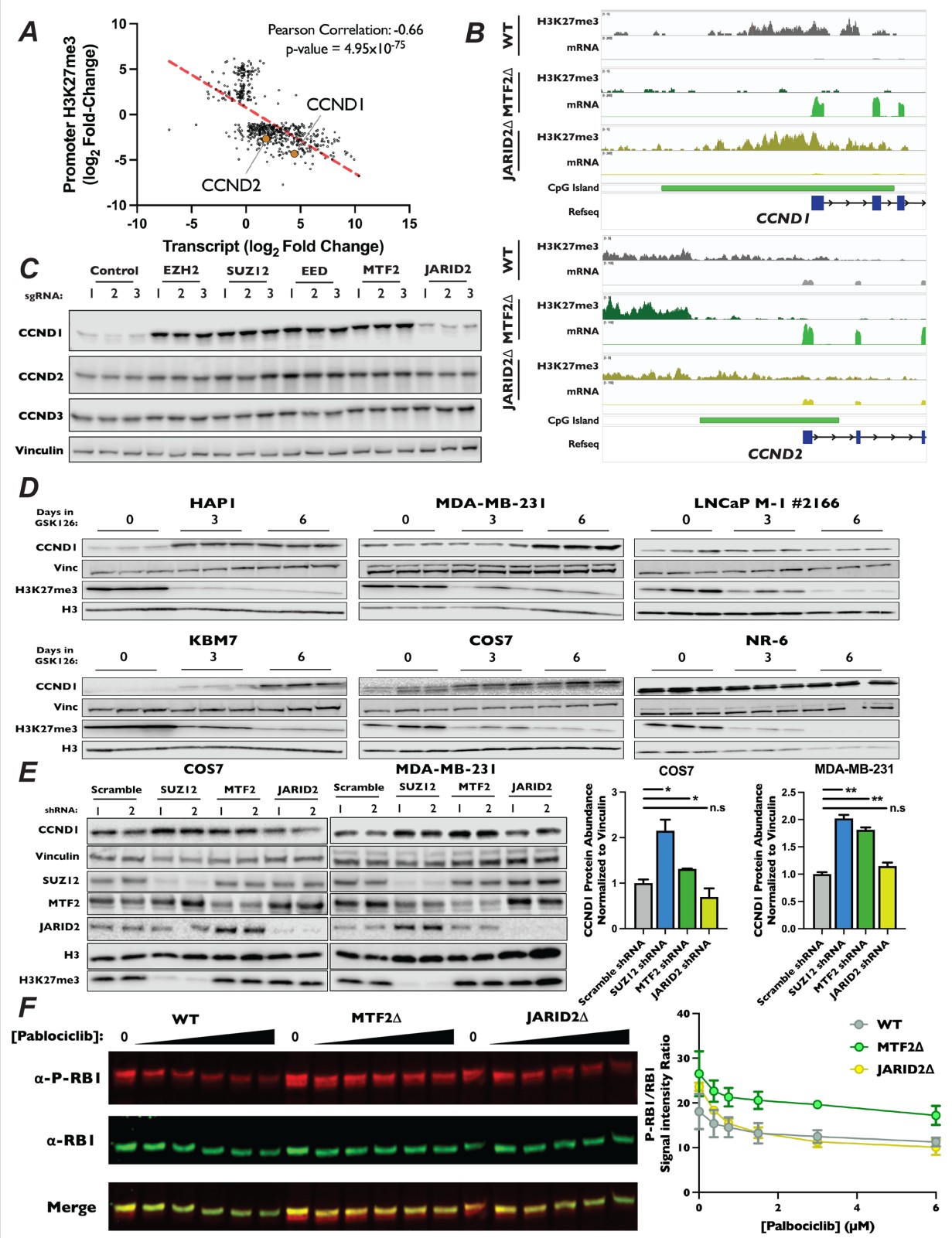

**Figure 6.** CCND1 and CCND2 expression is increased in MTF2Δ mutants. (**A**) Scatterplot of genes whose $\log_2$ fold-changes for MTF2Δ/wild-type ratio of mRNA expression (x-axis) versus promoter H3K27me3 signal (y-axis) had an adjusted p-value of <0.05 and an adjusted p-value <0.1 where plotted. (**B**) Genome browser traces of H3K27me3 signal, transcript abundance and CGIs in the CCND1 and CCND2 promoters. Annotated CGIs indicated by green bar. (**C**) *Top* – Western blots of Cas9-expressing pools of cells transduced three independent sgRNAs targeting the indicated genes, probed with

*Figure 6 continued on next page*

*Figure 6 continued*

the indicated antibodies. (**D**) Western blots of whole-cell lysates from a panel of cell lines treated with 10 μM GSK126 for the indicated time points, with listed antibodies. (**E**) *Left* – Western blots of whole-cell lysates from MDA-MB-231 and COS7 cells transduced with shRNA constructs shRNAs targeting SUZ12, MTF2, JARID2, or a scrambled control. Probed with indicated antibodies. *Right* – Quantification of western blots, CCND1 signal normalized to Vinculin. Each bar is the mean for two different shRNA expressing pools, error bars ±range. *p-value <0.05, **p-value <0.005, n.s.: not significant, two-tailed unpaired Student's *t*-test. (**F**) *Left* – Representative western blot of total RB1 and P-S807/8111-RB1 with increasing [palbociclib] in WT, MTF2Δ, and JARID2Δ cells, probed with indicated antibodies. *Right* – Quantification of the ratio of P-S807/8111-RB1 to total RB1 signal plotted against [palbociclib], two biological replicates, error bars ±range.

The online version of this article includes the following source data and figure supplement(s) for figure 6:

**Source data 1.** Original files for western blots shown in *Figure 6C–F*.

**Source data 2.** Original files for western blots shown in *Figure 6C–F*, indicating relevant band.

**Figure supplement 1.** Analysis of differential H3K27me3 distribution and transcript expression of D-type cyclins in MTF2Δ and JARID2Δ cell lines.

**Figure supplement 2.** Regulation of D-type cyclin expression by MTF2-containing PRC2.1 versus PRC2.2.

**Figure supplement 2—source data 1.** Original files for western blots shown in *Figure 6—figure supplement 2B*.

**Figure supplement 2—source data 2.** Original files for western blots shown in *Figure 6—figure supplement 2B*, indicating relevant band.

**Figure supplement 3.** CCND1 and CCND2 display increased expression in MTF2Δ cells.

**Figure supplement 3—source data 1.** Original files for western blots shown in *Figure 6—figure supplement 3B*.

**Figure supplement 3—source data 2.** Original files for western blots shown in *Figure 6—figure supplement 3B* indicating relevant band.

**Figure supplement 4.** Increased expression of CCND1 and CCND2 results in resistance to palbociclib.

**Figure supplement 4—source data 1.** Original files for western blots shown in *Figure 6—figure supplement 4B*.

**Figure supplement 4—source data 2.** Original files for western blots shown in *Figure 6—figure supplement 4B*, indicating relevant band.

CCND1 and CCND2 were the only upregulated canonical CDK4/6-RB-E2F pathway regulators in MTF2Δ cells.

Proper regulation of D-type cyclin expression is necessary to promote G1 progression, cellular fate specification, and organismal development, while dysregulation is seen in many cancers. Given the clear role of MTF2-dependent regulation of CCND1 and CCND2 in HAP1 cells, we sought to determine whether MTF2-containing PRC2.1 regulates the expression of D-type cyclins in other cell types in addition to HAP1. To probe this question, we determined levels of CCND1 by western blot in a panel of cell lines from a diversity of cell lineages treated with the EZH2 inhibitor GSK126 for 6 days. As expected, treatment of cells for 6 days with GSK126 resulted in a profound reduction in H3K27me3 levels in all cell lines tested and increased CCND1 expression in HAP1 cells (*Figure 6D*). Interestingly, in the non-adherent, haploid CML cell line KBM7, from which HAP1 was derived (*Carette et al., 2011*), we also observed a dramatic increase in CCND1 levels. This suggests that the regulation of D-type cyclin by PRC2 observed in HAP1 was not the results of artifacts introduced during the initial isolation of this cell line. Additionally, we observed that GSK126 treatment resulted in increased levels of CCND1 in the breast cancer line MDA-MB-231 and the SV40-immortalized monkey kidney fibroblast cell line COS7 (*Figure 6D*), suggesting that PRC2 activity can repress CCND1 expression in multiple mammalian cell lineages and species. In agreement with the context-dependent nature of PRC2 regulation of developmentally important loci, increased levels of CCND1 was not observed in the prostate adenocarcinoma cell line LNCaP and the immortalized mouse embryonic cell line NR-6, illustrating that regulation of CCND1 expression by PRC2 is not observed in all cellular lineages. To specifically probe the roles of PRC2.1 and PRC2.2 in the cell lines that displayed upregulation of CCND1 when all PRC2 activity was inhibited, we transduced COS7 and MDA-MB-231 cells with shRNAs targeting SUZ12, MTF2, and JARID2 and propagated cells for 2 weeks post selection. Consistent with results from propagation of these cell lines in GSK126, depletion of either SUZ12 and MTF2 in both COS7 and MDA-MB-231 resulted in significantly increased CCND1 expression (*Figure 6E*), suggesting that MTF2-containing PRC2.1 represses this loci in diverse cell types in addition to HAP1.

While D-type cyclins are necessary to promote the kinase activity of CDK4/6, they have also been shown to play roles outside of the RB1-E2F pathway (*Jirawatnotai et al., 2012*; *Shimura et al., 2013*; *Aggarwal et al., 2007*). Therefore, we sought to test if the increases in D-type cyclins seen in MTF2Δ cells lead to increased CDK4/6 activity, driving resistance to palbociclib treatment. First, we introduced alleles of CCND1 or CCND2 under the control of a doxycycline-inducible promoter into wild-type

HAP1 cells. We then performed a competitive proliferation assay in the presence of palbociclib and monitored the advantage conferred by overexpression of D-type cyclins. Consistent with the interpretation that the increased levels of D-type cyclins resulted in palbociclib resistance observed in MTF2Δ cells, overexpression of either CCND1 or CCND2 was sufficient to induce resistance to palbociclib (*Figure 6—figure supplement 4A*). Critically, as elevated cellular levels of D-type cyclins alone would be insufficient to drive palbociclib resistance, we tested to see if the increased expression of D-type cyclins in PRC2.1 mutant cell lines drove CDK4/6 association. To test this, we introduced a FLAG epitope-tagged copy of CDK6 into our knockout mutant cell lines, immunoprecipitated CDK6 and blotted for CCND1. We observed increased association of CCND1 with FLAG-tagged CDK6 in both SUZ12Δ and MTF2Δ, but not JARID2Δ, knockout cells (*Figure 6—figure supplement 4B*), suggesting that elevated levels of D-type cyclins indeed drive the formation of active CDK4/6 complexes. Finally, if the increased cellular levels of CCND1 and CCND2 seen in the MTF2Δ cell lines enhanced CDK4/6 kinase activity. To probe this question, we determined the extent of RB1 phosphorylation at S807 and S811, two well-characterized CDK4/6 targeted residues that are common markers of proliferation and which are recognized by a single antibody. To do this, we titrated wild-type, MTF2Δ, and JARID2Δ cells with increasing amounts of palbociclib and determined the levels of total RB1 and phosphorylated RB1 levels to calculate the ratio at each concentration. In each of our cell lines, higher concentrations of palbociclib resulted in decreased levels of phosphorylated RB1, as expected. However, compared to WT or JARID2Δ cells, MTF2Δ mutant cells maintained a higher ratio of phosphorylated to unphosphorylated RB1 at each concentration of palbociclib tested (*Figure 6F*). This result suggests that the increased levels of CCND1 and CCND2 in MTF2Δ cells increases CDK4/6 kinase activity, driving cells into S-phase (*Figure 4G*). In total, our results suggest that MTF2-contiaing PRC2.1 antagonizes G1 progression by repressing expression of the D-type cyclins CCND1 and CCND2 in certain cellular contexts.

## Discussion

Regulated progression through cell cycle phases is critical to normal cellular function and viability, while disordered progression is the hallmark of many disease states. Although the cell cycle has been an area of active research for decades, our understanding of its regulation remains incomplete. Using a chemogenetic approach, we found that inactivation of members of PRC2.1, but not factors specific to PRC2.2, resulted in profound resistance to the CDK4/6 inhibitor palbociclib. Loss of PRC2.1 complex members led to upregulation of the D-type cyclins CCND1 and CCND2, resulting in increased RB1 phosphorylation and S-phase entry in palbociclib-treated cells. We propose that PRC2.1, but not PRC2.2, mediates H3K27me3 deposition in the promoters CCND1 and CCND2 through the recognition of the CpG islands. These results tie PRC2.1 directly to the regulation of G1 progression.

   In the chemogenomic screens reported here, we recovered genes in a diverse array of biological pathways that resulted in sensitivity or resistance to well-characterized cell cycle inhibitors. In addition, we observed that inactivation of genes involved in mitochondrial homeostasis resulted in resistance to palbociclib. Small molecule inhibitors of EZH2 or the electron transport chain co-administered with palbociclib resulted in enhanced cell proliferation (*Figures 3D and 4D*), supporting the observed chemical–genetic interaction seen in our screen. However, genes identified in genetic screens should be interpreted with caution. Reproducible, and sometimes robust interactions can sometimes result from complicated changes in doubling time or alterations to the physiologic state of the cell (*Rahman et al., 2021*). It was recently demonstrated that genes encoding members of the electron transport chain are over-represented in DepMap co-dependency data, due to the remarkable stability of these protein complexes, which results in phenotypic lag that can vary in different backgrounds (*Rahman et al., 2021*). While mitochondrial complex assembly factors as enriched in Metascape analysis of our camptothecin screen as well as in palbociclib (*Figure 2C*), the enrichment was greater than 1600-fold more significant in palbociclib (*Figure 3A*). Moreover, a number of reports have found increased oxygen consumption and ROS production due to greater number and size of mitochondria in cells treated palbociclib (*Uzhachenko et al., 2021*; *Franco et al., 2016*; *Santiappillai et al., 2021*). This is consistent with a direct effect of CDK4/6 activity on mitochondrial function. Thus, in the case of both the PRC2 and the mitochondrial gene cluster, our data and that of others suggest that these results represent a direct link between these pathways and CDK4/6 biology.

Recently, PRC2 subcomplex accessory proteins have been implicated in an increasing number of processes that define cellular identity, including stem cell maintenance, differentiation, and cancer (*Rothberg et al., 2018*; *Ngubo et al., 2023*; *Deng et al., 2018*; *Parreno et al., 2022*). Despite the importance of controlled cellular division to each one of these processes, few reports have interrogated the roles of the different subcomplexes outside of stem cell model systems or specifically on their role in cell cycle regulation. Here, we show that in cells that lose either MTF2 or SUZ12 continue to proliferate despite palbociclib blockade (*Figure 4E, G*). These mutants show no apparent change in the proportion of cells undergoing apoptosis and display a greater proportion of cells entering S-phase in the presence of palbociclib, compared to wild-type or JARID2Δ cells. This increase is consistent with our findings that in MTF2Δ cell lines treated with palbociclib, a higher percentage of RB1 remains phosphorylated, while a similar increase is not seen in JARID2Δ cells. Consistently, the increased expression of CCND1 observed in PRC2.1 mutant cell results in increased the amounts of CCND1 associated with FLAG-tagged CDK6 (*Figure 6—figure supplement 4B*) and overexpression of either CCND1 or CCND2 was sufficient to drive palbociclib resistance in wild-type cells (*Figure 6—figure supplement 4A*). We surmise that the upregulation of CCND1 and CCND2 expression observed in cells lacking MTF2 results in increased CDK4/6 kinase activity that is sufficient to overcome palbociclib-mediated inhibition. Critically, we did not observe any significant changes in expression of other classic regulators of the CDK4/6-RB1-E2F pathway in either our CUT&RUN or RNA-Seq datasets (*Figure 6—figure supplement 3A, B*). While we cannot exclude the possibility that MTF2 inactivation alters the expression of other factors that influence G1 progression, we propose CCND1 and CCND2 represent major targets of PRC2.1 repression restraining G1 progression in HAP1 cells. Our results that MTF2 represses CCND1 expression in MDA-MB-231 (*Figure 6E*) may help explain the recent reports that low MTF2 expression leads to increased chemotherapeutic resistance in leukemia (*Maganti et al., 2018*) and downregulation of MTF2 was correlated with poorer clinical outcomes in breast cancer (*Liang et al., 2018*). However, more work is needed to determine whether D-type cyclins are critical PRC2.1 targets in tumors.

Work over the past decade has implicated accessory proteins as critical for proper genomic localization of the PRC2 enzymatic core. However, reports differ on in what chromatin and cellular contexts these subcomplexes act. Data from both mouse and human ES cells have suggested that PRC2.1 and PRC2.2 have overlapping genomic occupancy (*Healy et al., 2019*; *Youmans et al., 2021*), and that either subcomplex alone is capable of maintaining pluripotency (*Healy et al., 2019*; *Youmans et al., 2021*; *Højfeldt et al., 2019*). However, recent reports have found differing dependencies on these subcomplexes for proper distribution of H3K27me3 in cellular models of differentiation (*Glancy et al., 2023*; *Petracovici and Bonasio, 2021*; *Walker et al., 2010*; *Landeira et al., 2010*; *Pasini et al., 2010*). For example, a recent study in a model of induced differentiation suggested that MTF2 is involved in the maintenance of repression of PRC2 genes, whereas JARID2 is important for de novo deposition of H3K27me3 critical for gene silencing through genes 'pre-marked' with H2AK119ub1 (*Petracovici and Bonasio, 2021*) Conversely, PRC2.1 was shown to be required for the majority of H3K27me3 deposition during induced cell-fate transitions in mESCs, whereas PRC2.2 was not (*Glancy et al., 2023*). This study generated a triple knockout of all three PCL proteins (PHF1, MTF2, and PHF19), resulting in complete ablation of all PRC2.1 activity and did not probe the contribution of each accessory protein individually. Furthermore, MTF2 transcript levels are downregulated upon differentiation, whereas PHF1 and PHF19 levels increase (*Kloet et al., 2016*). These data suggest that the subunit composition of PRC2.1 changes during this process. In our experiments, MTF2 is the only PCL subunit important for D-type cyclin repression. These data, along with our results using the EZH2 inhibitor GSK126 (*Figure 6D*), are consistent with cell type-specific contributions of this class of proteins.

Using mutants of genes encoding subunits specific to either PRC2.1 or PRC2.2, we investigated the role of each subcomplex in cell cycle progression in HAP1 cells. In contrast to what has been demonstrated for ES cell lines where the two subcomplexes work synergically at the majority of sites (*Healy et al., 2019*; *Youmans et al., 2021*), we show that MTF2 is required for the majority of H3K27me3 deposition at CpG islands genome-wide and JARID2 was only partially required for H3K27me3 at these loci (*Figure 5E*). Importantly, the presence of MTF2 is more critical than JARID2 for the accumulation of H3K27me3 directly upstream of annotated transcription start sites in CpG islands-containing promoters in HAP1 cells (*Figure 5F*). Concordant with the patterns in H3K27me3 in promoters, we found that MTF2 loss resulted in a greater number of upregulated transcripts than

JARID2 loss (*Figure 5B*). Finally, MTF2Δ cells displayed a stronger correlation between genes with decreased promoter H3K27me3 levels and increased transcription than did JARID2Δ lines (*Figure 6A*, *Figure 6—figure supplement 1E*). However, we cannot exclude the possibility that AEBP2 plays a larger role in the activity of PRC2.2 than does JARID2 in these cells, as we identified AEBP2 as significantly, albeit modestly, increasing sensitivity to palbociclib in pooled knockout cells (*Figure 4A, E*). As H3K27me3 peak distribution was altered in the JARID2Δ cell lines (*Figure 5—figure supplement 2*), loss of JARID2 could alter H3K27me3 sites distal to promoters to change chromosome architecture or enhancer–promoter interactions. Alternatively, genes upregulated by loss of either MTF2 or JARID2 which did not have a significant alteration in promoter H3K27 methylation could be indirect effects. A recent report found that while PRC2.2 activity was not required for establishment of H3K27me3 during differentiation, but was instead required for recruitment of a PRC1 complex required for higher level chromatin interactions (*Glancy et al., 2023*). Future studies will be necessary to fully understand the coordination between these complexes.

The efficacy of CDK4/6 inhibitors in the treatment of HR+/HER2− breast cancer demonstrates the success of applying basic knowledge of cell cycle regulation to the generation of clinically relevant drugs. However, despite this success in the treatment of breast cancer, the efficacy of CDK4/6 inhibition is variable, with 10–20% of tumors primarily resistant and an additional 40% becoming resistant to these drugs within the first 2 years (*Johnston et al., 2019*; *Hortobagyi et al., 2016*). Moreover, CDK4/6 inhibitors are currently being explored for other tumor types, and these are each likely to have novel resistance mechanisms (*Parreno et al., 2022*; *Klein et al., 2018*; *Knudsen and Witkiewicz, 2017*). Thus, understanding perturbations in molecular pathways that can result in resistance to CDK4/6 inhibition could lead to improved patient responses and outcomes. In this study, we found that mutation of the PRC2.1 accessory protein MTF2 results in the development of resistance to palbociclib-induced proliferation reduction. Previously, EZH2, SUZ12, EED, MTF2, and JARID2 have all been suggested to not only act as oncogenes (*Béguelin et al., 2013*; *Zhao et al., 2021*; *McCabe et al., 2012*; *Li et al., 2012*; *Wu et al., 2018*; *Liu et al., 2015*; *Wang et al., 2020*; *Wu et al., 2019*), but also to have tumor suppressor activities (*Su et al., 2015*; *Jadhav et al., 2020*; *Deng et al., 2018*; *Maganti et al., 2018*; *Liang et al., 2018*; *Ntziachristos et al., 2012*; *Mieczkowska et al., 2021*), depending on the type of cancer. Additionally, other chemogenetic screens utilizing palbociclib and have not identified that inactivation of PRC2 components as either enhancing or reducing palbociclib-induced proliferation defects (*Chaikovsky et al., 2021*; *Poulet et al., 2024*), suggesting that PRC2 mutation is neutral in the cell lines studied. These observations not only underscore the context-dependent ramifications of mutation of these PRC2 complex members, but also may help inform the context in which CDK4/6 inhibitors are most efficacious. Clinical trials using CDK4/6 inhibitors in combination with other therapeutics are underway and the mutational status and expression levels of PRC2 subunits might serve as predictors of efficacy.

## Lead contact and materials availability

Requests for further information and reagents should be directed to and will be fulfilled by the Lead Contact, David Toczyski (dpt4darwin@gmail.com).

## Methods
### Cell lines

Cas9-expressing HAP1 and KBM7 cells were cultured in IMDM (Gibco) supplemented with 4 mM glutamine (Gibco), 10% Tetracycline-free FBS (Sigma-Aldrich) and either 1× Antibiotic, Antimycotic (Invitrogen) or 1% penicillin–streptomycin (Sigma-Aldrich). HAP1 cells stably expressed Cas9 were employed for the whole-genome screen, while for subsequent experiments, an HAP1 line harboring a doxycycline-inducible Cas9 was utilized. COS7, MDA-MB-231, NR-6, and HEK293T cells used for the production of virus were cultured in DMEM (Gibco) supplemented with 2mM glutamine (Gibco), 10% Tetracycline-free FBS (Sigma-Aldrich) in 1× Antibiotic, Antimycotic (Invitrogen). LNCaP M-1 #2166 cells were cultured in RPMI-1640 (Gibco) supplemented with 2 mM glutamine (Gibco), 10% Tetracycline-free FBS (Sigma-Aldrich) in 1× Antibiotic, Antimycotic (Invitrogen). Cells were detached from tissue culture dishes using 0.25% Trypsin (Gibco) and maintained at 37°C, 5% $CO_2$. Our laboratory conducts

regular mycoplasma testing of cultured cells with the MycoAlert Mycoplasma Detection kit (Lonza), and no mycoplasma contamination of any cell line was detected during this study.

## Genome-wide chemical screening

The lentiviral TKOv3 sgRNA library (Addgene #90294) was used to perform pooled genome-wide CRISPR knockout screens. The library contains 70,948 guides, targeting 18,053 protein-coding genes (4 guides/gene). Ninety million HAP1 cells stably expressing Cas9 were seeded into 15 cm dishes and infected with TKOv3 lentivirus at a multiplicity of infection of roughly 0.3, such that every sgRNA is represented in approximately 200–300 cells after selection (>200-fold coverage). After 24 hr of infection, cells with successful viral integration were selected in 25 ml IMDM medium containing 1 µg/ml puromycin (Sigma-Aldrich). Selection took place for 48 hr. Following selection, cells were harvested, pooled, and split into three replicates of 15 million cells each to maintain >200-fold coverage of the sgRNA library (day 0). At day 3, each replicate was split such that every drug screen had a at least 15 million cells per replicate to maintain >200-fold coverage. The drug concentrations ($IC_{30}$–$IC_{50}$ determined as described below) used in the genome-wide chemical screens were as follows: palbociclib – 0.7 µM, colchicine – 9.2 nM, and camptothecin – 1 nM. An increase in potency was observed for most drugs when used in the pooled screens, thus screening concentrations were adjusted to preserve $IC_{30}$–$IC_{50}$ throughout each passage. Cells were subject to treatment with drug in 0.1% DMSO, or 0.1% DMSO alone. Drug-containing media was refreshed every 3 days, along with the passaging of cells and the collection of cell pellets. To preserve >200-fold coverage, 20 million cells were pelleted with every passage, from day 0 to 18.

Genomic DNA extraction and sequencing library preparation were performed as described previously (*Aregger et al., 2020*). Briefly, genomic DNA from cell pellets were extracted using the Wizard Genomics DNA Purification Kit (Promega) and quantified using the Qubit dsDNA Broad Range Assay kit (Invitrogen). Sequencing libraries were prepared as described previously (*Michael et al., 2019*). Briefly, two PCR amplification steps were performed to first enrich for the sgRNA regions in the genome and second, attach Illumina sequencing indices to the amplified regions. Sequencing libraries were prepared from 50 µg of genomic DNA (200-fold library coverage) using the NEBNext Ultra II Q5 Polymerase (NEB). Primers used included Illumina TruSeq adapters with i5 and i7 indices. Barcoded libraries were gel-purified using the PureLink Quick Gel Extract kit (Thermo Fisher) and sequenced on an Illumina HiSeq2500.

## Drug concentrations for chemical screening

Drug dosing experiments were performed to determine screening concentrations. HAP1 cells stably expressing Cas9 were seeded at a density of 2.5 million cells per 15 cm dish. Cells were treated with 0.1% DMSO, or drug in 0.1% DMSO, 2 hr after seeding. Viable adherent cells were counted 2 days post-treatment on a Coulter counter, and inhibitory concentrations were determined. The following are ranges of drug concentrations used in the dosing experiments: palbociclib: 1.5–10 µM, colchicine: 1.5–150 nM, and camptothecin: 1–5 nM.

## Orobas pipeline for scoring chemical genetic interactions

The Orobas pipeline (version 0.5.0) was used to score chemical genetic interactions from the genome-wide CRISPR/Cas9 screen data. The process is summarized here, and the complete R code is provided as a source code file. sgRNAs were normalized to sequencing depth for each sample and the LFC in sgRNA abundance was calculated for each condition relative to the corresponding T0 sample. Guides with fewer than 30 read counts in the T0 sample were filtered out from further analysis, and genes with fewer than threeremaining guides post-filtering were also filtered out from scoring. Residual effects were computed for each gene by calculating the residual LFC between sgRNAs in treated versus DMSO samples after averaging technical replicate LFCs. Residual effects were then M-A transformed and Loess-normalized to account for potential skew and non-linearity present in the data, and per-gene effect sizes and FDRs were computed by applying the moderated *t*-test to normalized residual effects. Hits were called as genes with FDRs less than 0.4 and per-gene effect sizes greater than 0.5 or less than −0.5 (a complete list of effect sizes and FDRs is included as *Supplementary file 1*).

## STRING interaction network generation

STRING networks were set to only display physical interactions scores that were returned with high confidence (0.7) and taken from text-mining, experiments, and databases.

## sgRNA lentiviral vector cloning

Oligos for sgRNA targets were designed to contain the 5′ overhang CACCG- for the sense oligo 5′ and for that antisense 3′ over hang AAAC- and -C, respectively. 10 µM each of sense and antisense oligos (Integrated DNA Technologies) were mixed in 1× T4 DNA Ligase buffer and water to a total volume of 10 µl. This mixture was heated to 95°C for 5 min, then oligos were annealed by decreasing the temperature at a rate of −0.1°C/s till the mix reached 25°C. Annealing reactions were diluted 1:10 with water and then 1 µl was used to ligate into 100 ng of BsmBI digested pLentiGuidePuro vector (Addgene #52963) in 1× T4 DNA Ligase Buffer. 600 units of T4 DNA Ligase (NEB) and water to a total volume of 25 µl. After incubating for 1hr at 37°C, 2 µl of the ligation reaction was transformed into β-ME pre-treated XL10-Gold cells (Agilent) per the manufacturer's instructions and plated on LB + 100 µg/ml carbenicillin plates for selection. Plasmids recovered from single colonies were confirmed by Sanger sequencing.

## Polyclonal and monoclonal knockout generation

Cas9-expressing HAP1 cells were transduced with pLentiGuidePuro vectors (Addgene #52963) expressing a single sgRNA (see *Supplementary file 5* for sgRNA sequence). Lentiviral transduction was conducted at low MOI (~30%) following standard protocols. Integration of the sgRNA was selected with 1 µg/ml puromycin for up to 2 days, followed by combined puromycin selection and Cas9 induction for 3 days with 1 µg/ml doxycycline. This polyclonal pool of pLentiGuidePuro transduced cells was then used for 'pooled' knockout experiments or used to generate monoclonal cell lines. Trypsinized, single cells were then sorted into individual wells in a 96-well plate using the Sony SH800 sorter (UCSF, LCA). Isolated single-cell-derived colonies were screened for mutation by PCR, followed by Sanger sequencing of the purified PCR product and ICE analysis (Synthego) of the resulting chromatographs. Candidate clonal knockouts were then confirmed by western blot. Only monoclonal lines that clearly displayed knockout alleles and had no protein product by western blot were utilized further.

## Competitive growth assays (GFP/BFP pooled knockouts and GFP$^+$/monoclonal knockout pools)

For pooled knockout competitive growth assays, HAP1 cells harboring an inducible Cas9 and expressing GFP and HAP1 cells expressing BFP (Hundley et al.) were mixed at a ratio of 1:4 GFP:BFP HAP1 cells into a single well, with three GFP/BFP cell mixtures for each gene targeted for inactivation. Mixtures were transduced at a low MOI with a pLentiGuidePuro vector expressing one sgRNA (three biological replicates per gene, sgRNAs in *Supplementary file 5*). After 24 hr of lentiviral transduction, pools of cells were selected with 1 µg/ml puromycin for 1 day, followed by 1 µg/ml puromycin and doxycycline for 3 days to select for sgRNA integration and to induce Cas9 expression. After 3 days of Cas9 induction, pools were split into media with or without palbociclib every 3 days, for 18 days. The GFP/BFP ratio was monitored on the Attune NxT (Invitrogen) flow cytometer every 3 days. FlowJo v10 was used to determine the GFP/BFP ratio at each time point. The ratio of GFP to BFP was normalized to the day 0 ratio (prior to splitting into palbociclib), and subsequently to the matched untreated ratio at each time point.

For HAP1 GFP$^+$/GFP$^-$ competitive growth assays, wild-type, doxycycline-inducible CCND1, doxycycline-inducible CCND2, clonal MTF2Δ, or JARID2Δ GFP$^-$ cells were mixed with HAP1 wild-type mGFP$^+$ clones at 1:5 GFP$^-$:GFP$^+$ ratio, split into media with or without drug, and analyzed by flow cytometry as described above.

## Western blotting

Harvested cell pellets were lysed in 1× RIPA buffer supplemented with 1× EDTA-free cOmplete protease inhibitor (Roche) and 1× PhosphoSTOP phosphatase inhibitor (Roche) for 30 min on ice with two rounds of 15-s vortexing. Lysates were cleared at 21,000 × *g* for 10 min at 4°C. Protein concentration was determined by BCA assay and BSA standard curve (Pierce), and samples were adjusted to 1 µg/µl total protein with 1× RIPA and SDS–PAGE sample loading buffer was added (62.5 mM Tris-HCl

(pH 6.8), 2.5% SDS, 0.1% bromophenol blue, 10% glycerol, 5% β-mercaptoethanol (vol/vol)). 10 µg of total protein was loaded per lane onto a 4–20% Criterion Tris-HCl Protein gel (Bio-Rad) and separated by electrophoresis at 150 V for 1 hr. Proteins were transferred and immobilized onto a nitrocellulose membrane (GE Healthcare) by electrophoresis for 1 hr at 100 V in standard transfer buffer containing 20% methanol. Membranes were blocked for an hour at room temperature and then then probed overnight in a 1:1000 dilution of 1° antibody (unless otherwise indicated) at 4°C and in a 1:10,000 2° antibody at room temperature for 1 hr at in the appropriate blocking buffer. Chemiluminescent and fluorescent signals were visualized with an Odyssey FC imager (LICOR).

## Cell cycle analysis by propidium iodine

200,000 cells/well were plated in 6-well dishes at, as to be 10–20% confluent at the time of treatment. Cells were treated with inhibitors 24 hr after plating, then harvested 48 hr later by trypsinization, washed twice with cold 1× PBS, fixed by dropwise addition of ice-cold 70% ethanol, and incubated at 4°C overnight. Fixed samples were washed twice with 1× PBS + 1% BSA prior to resuspension in a solution of 1× PBS, 1 mg/ml RNase A and 50 µg/ml propidium iodide for 1 hr at 37°C. DNA content of at least 20,000 single cells was determined by Attune NxT flow cytometer (Invitrogen), and data were analyzed using FlowJo v10.

## BrdU incorporation assay

250,000 cells/well were plated in 6-well dishes and grown for 24 hr prior to treatment. Cells were then treated with either DMSO (mock) or 1.5 µM palbociclib for a total of 24 hr, with 10 µM BrdU being added to the culture medium 1 hr prior to harvesting. Cells were counted using the Countess automatic hemocytometer (Invitrogen) to ensure that only 1 million cells were stained. Cells were prepared for analysis using BD Pharmagen BrdU Flow Kits (BD Biosciences) according to the manufacturer's instructions. BrdU incorporation was determined for at least 20,000 single cells by Attune NxT flow cytometer (Invitrogen), and data were analyzed using FlowJo v10.

## Quantitative Crystal Violet proliferation assay

1 ml of a 1000 cells/ml suspension were seeded into a per well in a 6-well plate containing 1 ml IMDM supplemented with double the indicated concentration of palbociclib and GSK126 in technical triplicate. Cells were allowed to proliferate for 9 days, with the media supplemented with the drug at the concentration indicated replaced every 3 days. After 9 days, cells were washed once with 1× PBS, followed by staining and fixation in a 0.25% Crystal Violet, 20% methanol solution for 10 min at room temperature. Following staining, cells were washed six times with 1× PBS and lysed in a 100 mM sodium citrate and 50% ethanol solution for 30 min at room temperature on an orbital shaker. Lysates were recovered and absorbance at 590 nM was detected using a Synergy Neo2 Microplate Reader (BioTek). Proliferation at each concentration was determined relative to untreated wells.

## PrestoBlue proliferation assay

45 µl of a 50,000 cells/ml cell suspension was seeded into a 96-well plate containing 45 µl of IMDM supplemented with the indicated concentration of palbociclib, antimycin A, TTFA or oligomycin in triplicate. After proliferation for 48 hr, 10 µl of PrestoBlue (Invitrogen) was added to each well and incubated for 30 min at 37°C. Conversion of PrestoBlue was determined by recording the fluorescence excitation at 560 nM and emission at 590 nM using a Synergy Neo2 Microplate Reader (BioTek). Proliferation at each concentration was determined relative to untreated wells.

## RNA extraction

150,000 cells were seeded into 6-well plates and allowed to grow overnight. Cells were treated with DMSO (Mock) or 1.5 µM palbociclib for 24 hr prior to harvesting directly in TRIzol reagent (Invitrogen). After chloroform extraction, the aqueous phase was transferred to a fresh tube and 1 volume of 100% ethanol was added and mixed thoroughly before binding to an RNA Clean & Concentrator (Zymo). RNA was DNase I digested on-column (Zymo), purified according to the manufacturer's instructions and eluted in nuclease-free water. To prepare RNA-Seq libraries, 2 µg of total RNA was polyA, followed by Illimina adaptor ligation and paired-end sequencing on an Illumina HiSeq at a depth of at least 22 million reads per sample by Azenta.

## First strand cDNA synthesis and qRT-PCR

2 μg of total RNA was first heat denatured in the presence of dNTPs and oligo-dT at 65°C for 5 min. RNase inhibitor and Tetro reverse transcriptase (Bioline) was then added to heat denatured total RNA and cDNA was synthesized at 45°C for 1 hr, followed by heat inactivation at 85°C for 5 min. cDNA synthesis reactions were then diluted 1:5 and 2 μl was added into qRT-PCR reaction mix, utilizing SensiFast Lo-ROX qRT-PCR Mastermix (Bioline) in both biological and technical triplicate. Reactions were carried out and analyzed using a QuantStudio5 machine (Applied Biosystems). See *Supplementary file 5* for qRT-PCR primer sequences.

## FLAG immunoprecipitation

2xFLAG-2xStrep-CDK6 expressing PRC2 component monoclonal knockout mutant HAP1 cells were harvested and lysed on ice in a 10 mM HEPES, 150 mM NaCl, 1 mM $MgCl_2$, 1 mM EDTA, 1% NP-40 lysis buffer supplemented with cOmplete Protease Inhibitor (Roche), and PhosphoSTOP (Roche) tablets by passage through an 21-gauge syringe and rotation at 4°C for 30 min. Lysates were clarified by centrifugation at 21,000 rcf for 15 min at 4°C and the protein content was quantified and the input was normalized to 5 μg/immunoprecipitation. 150 μl of FLAG-conjugated Dynabeads slurry (Invitrogen) was added to each normalized lysate and immunoprecipitated overnight at 4°C under constant rotation. The following immunoprecipitation, Dynabeads were washed five times with a 1× PBS, 0.1% NP-40 buffer and bound protein was eluted from the beads in 1× PBS, 0.1% NP-40 buffer supplemented with 5 μg of 3x-FLAG peptide (Sigma). Resulting eluates were then analyzed by SDS–PAGE and western blots.

## CUT&RUN library preparation

CUT&RUN libraries were generated by first lysing 300,000–500,000 cells in 500 μl of Nuclei Extraction Buffer (20 mM HEPES–KOH pH 7.9, 10 mM KCl, 1 mM $MgCl_2$, 0.1% Triton X-100, 20% glycerol, and 1× protease inhibitor) for 10 min on ice. Next, samples were spun down and washed twice with Nuclei Extraction Buffer before being resuspended in 500 μl nuclei extraction buffer. 10 μl of Concanavalin A-coated beads (EpiCypher) previously washed in Wash Buffer (20 mM HEPES–KOH pH 7.5, 150 mM NaCl, 2 mM EDTA, 0.5 mM spermidine, and 1× protease inhibitor) and resuspended in Binding Buffer (20 mM HEPES–KOH pH 7.5, 1 mM $CaCl_2$, and 1 mM $MnCl_2$) were then added to the samples and incubated with rotation for 15 min at 4°C. Next, samples were washed once with Binding Buffer before being resuspended in 50 μl of Buffer 2 containing 0.1% BSA, 2 μM EDTA, and 0.5 μl H3K27me3 1° antibody, followed by overnight incubation with rotation at 4°C. Following the incubation, samples were washed twice with Buffer 2 before being incubated in 50 μl of Buffer 2 containing ~700 ng/ml Protein A-MNase fusion protein (Batch #6 from the Henikoff Lab) for 1 hr with rotation at 4°C. Samples were washed two more times and resuspended in 100 μl of Buffer 2 before starting the MNase digestion by adding $CaCl_2$ to a concentration of 2 mM on ice for 30 min, after which the reaction was quenched with the addition of 100 μl 2× Stop Buffer (200 mM NaCl, 20 mM EDTA, 4 mM EGTA, 50 μg/ml RNase A, 40 μg/ml GlycoBlue (Ambion), and 2 pg/ml spike-in DNA) to inactivate the MNase. Samples were incubated for 30 min at 37°C and spun down for 5 min at 4°C to release DNA fragments. DNA was phenol:chloroform extracted and 200 μl of the recovered aqueous phase was ethanol precipitated with 500 μl ethanol, 20 μl 3 M NaOAc, 2 μl GlycoBlue at −80°C. Libraries were prepared using 2S Plus DNA Library Kit adapters (Swift Biosciences) and size-selected using SPRIselect beads (Beckman Coulter) before being amplified and sent for paired-end sequencing on the NovaSeq 6000 (150 bp reads).

## CUT&RUN processing and analysis

CUT&RUN paired reads were aligned to a reference human genome (hg38) by the bwa-mem algorithm. PCR duplicate reads were removed by Picardand peaks were called using macs2 with the broad flag and an FDR of 0.05. Bedtools intersect was used to identify reproducible peaks between biological replicates of each condition, and reproducible peaks from each condition were compiled into a list. Bedtools multicov was used to build a matrix with the number of reads from each dataset falling in each region in this list. This matrix was used for all 'genome-wide' analyses. Bedtools multicov was also used to build a matrix with the number of reads from each dataset in a 5000-bp window around the transcription start site (4 kb upstream, 1 kb downstream) of all hg38 genes defined by gencode

v41. The gencode v41annotation for CCND2 was originally incorrectly assigned to chr12:4,265,771–4,270,771 and reassigned using the Refseq coordinates chr12:4,269,762–4,274,762. Count matrices were analyzed with DESeq2 to compare changes in H3K27me3 deposition globally, and changes in H3K27me3 deposition in promoters. For heatmaps, deduplicated BAM files were converted to bigwigs and BED files and normalized reads per kilobase per million mapped read using deepTools bamCoverage. For genome-wide analyses, H3K27me3 CUT&RUN signal in normalized bigwigs was measured using deepTools computeMatrix in 10 kb regions centered around WT peaks overlapping with CpG islands. For promoter analyses, H3K27me3 CUT&RUN signal in normalized bigwigs was measured using deepTools computeMatrix in 5 kb regions (4 kb upstream, 1 kb downstream) around transcription start sites for promoters overlapping with CpG islands. Promoters with the highest average H3K27me3 signal intensity in wild-type replicate 1 and sorted in descending order.

## RNA-Seq processing and analysis

RNA-Seq paired reads were quantified using Salmon. Transcript-level abundance estimates from Salmon and gene-level count matrices were created using Tximport and analyzed using DESeq2. Paired reads were aligned using STAR to generate BAM files. BAM files were converted to BED files using bamCoverage and normalized using RPKMs and to effective genome size of hg38 (2,913,022,398) with a bin size of 10.

## Acknowledgements

I would like to thank the members of the Toczyski, Jura, Ramani, and Shen lab for their experimental and intellectual feedback on the preparation of this manuscript. I would specifically like to thank Hiten Madhani and Natalia Jura for their thoughtful discussions on this project and for mediating collaborations critical in completion of this work. Additionally, I would like to thank Nerea Sanvisens Delgado for initial compound dosing used in this study and Sarah Elmes for support in developing our flow cytometry experiments. This work was supported by an NIH grant R35 GM118104, a grant from the UCSF Breast Oncology Program (BOP) and the Sandler Program for Breakthroughs in Biomedical Research (PBBR) to David Toczyski and R01AG057497, R01EY027789, UM1HG009402, and U01DA052713 to Yin Shen. DNA sequencing was performed at the UCSF Center for Advanced Technology, and flow cytometry and qRT-PCT at UCSF Laboratory of Cell Analysis.

## Additional information

### Funding

| Funder | Grant reference number | Author |
| --- | --- | --- |
| NIH Office of the Director | R35 GM118104 | David P Toczyski |
| Sandler Program for Breakthroughs in Biomedical Research (PBBR) | R01AG057497 | Yin Shen |
| Sandler Program for Breakthroughs in Biomedical Research (PBBR) | R01EY027789 | Yin Shen |
| Sandler Program for Breakthroughs in Biomedical Research (PBBR) | UM1HG009402 | Yin Shen |
| Sandler Program for Breakthroughs in Biomedical Research (PBBR) | U01DA052713 | Yin Shen |

| Funder | Grant reference number | Author |
|--------|------------------------|--------|
| Sandler Program for Breakthroughs in Biomedical Research (PBBR) | | David P Toczyski |

The funders had no role in study design, data collection and interpretation, or the decision to submit the work for publication.

## Author contributions

Adam D Longhurst, Conceptualization, Formal analysis, Validation, Investigation, Visualization, Methodology, Writing – original draft, Writing – review and editing; Kyle Wang, Investigation, Methodology, Writing – review and editing, Conducted chemogenetic CRISPR-Cas9 whole-genome screen; Harsha Garadi Suresh, Investigation, Methodology, Conducted chemogenetic CRISPR-Cas9 whole-genome screen; Mythili Ketavarapu, Formal analysis, Visualization, Writing – review and editing; Henry N Ward, Software, Formal analysis, Visualization; Ian R Jones, Resources, Writing – review and editing, assisted in generating the CUT&RUN sequencing libraries; Vivek Narayan, Supervision, Assisted in generating the CUT&RUN sequencing libraries; Frances V Hundley, Methodology, Writing – review and editing; Arshia Zernab Hassan, Methodology, Project administration; Charles Boone, Chad L Myers, Vijay Ramani, Brenda J Andrews, Supervision, Methodology, Project administration; Yin Shen, Supervision, Project administration; David P Toczyski, Conceptualization, Funding acquisition, Writing – original draft, Project administration, Writing – review and editing

## Author ORCIDs

Adam D Longhurst http://orcid.org/0000-0002-2463-8370
Brenda J Andrews https://orcid.org/0000-0001-6427-6493
David P Toczyski https://orcid.org/0000-0001-5924-0365

Reviewer #1 (Public review): https://doi.org/10.7554/eLife.97577.3.sa1
Reviewer #2 (Public review): https://doi.org/10.7554/eLife.97577.3.sa2
Reviewer #3 (Public review): https://doi.org/10.7554/eLife.97577.3.sa3
Author response https://doi.org/10.7554/eLife.97577.3.sa4

# Additional files

## Supplementary files

Supplementary file 1. Orobas analysis of palbociclib, camptothecin, and colchicine chemogenetic screens.

Supplementary file 2. Genes identified as significantly enriched or de-enriched in palbociclib, camptothecin, and colchicine chemogenetic screens.

Supplementary file 3. DESeq2 results of promoter with significant H3K27me3 signal enrichment in CUT&RUN experiment.

Supplementary file 4. DESeq2 results of transcripts with significant changes in RNA-Seq experiment.

Supplementary file 5. List of cell lines, reagents, and oligonucleotides used in this study.

MDAR checklist

Source code 1. Orobas analysis pipeline source code.

## Data availability

Cell lines used in this study are listed in Supplementary File 5 and are available upon request. Sequencing data generated during the course of this are available using the NCBI accession number PRJNA1201849.

The following dataset was generated:

| Author(s) | Year | Dataset title | Dataset URL | Database and Identifier |
|---|---|---|---|---|
| University of California San Francisco | 2024 | The PRC2.1 Subcomplex Opposes G1 Progression through Regulation of CCND1 and CCND2 | https://www.ncbi.nlm.nih.gov/bioproject/PRJNA1201849/ | NCBI BioProject, PRJNA1201849 |

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
