## [Editor Report · eLife Assessment]

This **valuable** study reports a chemogenetic screen for resistance and sensitivity to three cell cycle inhibitors used in the clinic: camptothecin, colchicine, and palbociclib. The screen provides a wealth of information that will be of interest to cell cycle and cancer biologists. **Convincing** evidence is provided that resistance to palbociclib can result from loss of PRC2.1 activity, which raises cyclin D levels. The effect of PRC2.1 on cyclin D is not universal across tested cell lines with the causal differences not yet understood.

---

## [Referee Report · Reviewer #1 (Public review)]

The study by Longhurst et al. investigates the mechanisms of chemoresistance and chemosensitivity towards three compounds that inhibit cell cycle progression: camptothecin, colchicine, and palbociclib. Genome-wide genetic screens were conducted using the HAP1 Cas9 cell line, revealing compound-specific and shared pathways of resistance and sensitivity. The researchers then focused on novel mechanisms that confer resistance to palbociclib, identifying PRC2.1. Genetic and pharmacological disruption of PRC2.1 function, but not related PRC2.2, leads to resistance to palbociclib. The researchers then show that disruption of PRC2.1 function (for example, by MTF2 deletion), results in locus-specific changes in H3K27 methylation and increases in D-type cyclin expression. The study shows that increased expression of D-type cyclins results in palbociclib resistance.

Strengths:

The results of this study are interesting, and the study contributes insights into the molecular mechanisms of CDK4/6 inhibitors. Importantly, while CDK4/6 inhibitors are effective in the clinic, tumour recurrence is very high due to acquired resistance.

Weaknesses:

A key resistance mechanism is Rb loss, so it is important to understand if resistance conferred by PRC2.1 loss is mediated by Rb, and whether restoration of PRC2.1 function in Rb-deplete cells results in renewed palbociclib sensitivity. It is also important to understand the clinical implications of the results presented. Inclusion of these data would significantly improve the paper. At present, it is unclear if mutations in PRC2.1 are found in genetic analyses of tumour samples in patients with acquired resistance.

---

## [Referee Report · Reviewer #2 (Public review)]

Summary:

Longhurst et al. assessed cell cycle regulators using a chemogenetic CRISPR-Cas9 screen in the haploid human cell line HAP1. Besides known cell cycle regulators they identified the PRC2.1 subcomplex to be specifically involved in G1 progression, given that the absence of members of the complex makes the cells resistant to Palbociclib. They further showed that in HAP1 cells the PRC2.1, but not the PRC2.2 complex is important to repress the cyclins CCND1 and CCND2. This can explain the enhanced resistance to Palbociclib, a CDK4/6-Inhibitor, after PRC2.1 deletion.

Strengths:

The initial CRISPR screen is very interesting, because it uses three distinct chemicals that disturb the cell cycle at various stages. This screen mostly identified known cell cycle regulators, which demonstrates the validity of the approach. The results can be used as a resource for future research.

The most interesting outcome of the experiment is the finding that knockouts of the PRC2.1 complex make the cell resistant to Palbociclib. In further experiments, the authors focused on MTF2 and JARID2 as main components of PRC2.1 and PRC2.2, respectively. Via extensive analyses, including genome-wide experiments, they confirmed that MTF2 is particularly important to repress the cyclins CCND1 and CCND2. Absence of MTF2 therefore leads to increased expression of these genes, sufficient to make the cell resistant to Palbociclib. This result will likely be of wide interest to the community.

Weaknesses:

The work is limited to specific biological contexts, and the generality of the conclusions is uncertain.

Comments on revisions:

The revision offers new insights and is overall satisfying. I have no further recommendations that I consider essential.

---

## [Referee Report · Reviewer #3 (Public review)]

This study begins with a chemogenetic screen to discover previously unrecognized regulators of the cell cycle. Using a CRISPR-Cas9 library in HAP1 cells and an assay that scores cell fitness, the authors identify genes that sensitize or desensitize cells to the presence of palbociclib, colchicine, and camptothecin. The results suggest that these three drugs inhibit proliferation through different mechanisms, and with each treatment, expected and unexpected pathways were found to affect drug sensitivity. The authors focus the rest of the experiments and analysis on the polycomb complex PRC2, as deletion of several of its subunits in the screen conferred palbociclib resistance. The authors find that PRC2, specifically a complex dependent on the MTF2 subunit, methylates histone 3 lysine 27 (H3K27) in promoters of genes associated with various processes including cell-cycle control. Further experiments demonstrate that Cyclin D expression increases upon loss of PRC2 subunits, providing a potential mechanism for palbociclib resistance.

The strengths of the paper are the design and execution of the chemogenetic screen, which provides a wealth of potentially useful information. The data convincingly demonstrate in the HAP1 cell line that the MTF2-PRC2 complex sustains the effects of palbociclib (Fig. 4), methylates H3K27 in CpG-rich promoters (Fig. 5), and represses Cyclin D expression (Fig. 6). The correlation between MTF2-PRC2 inhibition and increased Cyclin D levels is shown in multiple cell lines using both genetic and chemical approaches. These results could be of great interest to those studying cell-cycle control, resistance mechanisms to therapeutic cell-cycle inhibitors, and chromatin regulation and gene expression.

There are a few weaknesses that somewhat temper the overall quality and potential impact of the study. First, the results from the colchicine and camptothecin screens (Fig. 1 and 2) are not experimentally validated, which lessens the rigor of those data and conclusions. Second, some experiments validating and further exploring results from the palbociclib screen (Figs. 4 and 5) are restricted to the Hap1 cell line, so the generality of some conclusions is not established. Third, conclusions drawn from data in Fig. 4D are not fully supported by proper use of biological replicates and analysis of the results.

Comments on revisions:

Proper statistical analysis considering biological replicates is still not applied to determine whether differences in palbociclib IC50 values at different GSK126 concentrations are significant.

---

## [Author Response]

The following is the authors’ response to the original reviews.

**Public Reviews:**

**Reviewer #1 (Public Review):**
The study by Longhurst et al. investigates the mechanisms of chemoresistance and chemosensitivity towards three compounds that inhibit cell cycle progression: camptothecin, colchicine, and palbociclib. Genome-wide genetic screens were conducted using the HAP1 Cas9 cell line, revealing compound-specific and shared pathways of resistance and sensitivity. The researchers then focused on novel mechanisms that confer resistance to palbociclib, identifying PRC2.1. Genetic and pharmacological disruption of PRC2.1 function, but not related PRC2.2, leads to resistance to palbociclib. The researchers then show that disruption of PRC2.1 function (for example, by MTF2 deletion), results in locus-specific changes in H3K27 methylation and increases in D-type cyclin expression. It is suggested that increased expression of D-type cyclins results in palbociclib resistance.Strengths:The results of this study are interesting and contribute insights into the molecular mechanisms of CDK4/6 inhibitors. Importantly, while CDK4/6 inhibitors are effective in the clinic, tumour recurrence is very high due to acquired resistance.Weaknesses:A key resistance mechanism is Rb loss, so it is important to understand if resistance conferred by PRC2.1 loss is mediated by Rb, and whether restoration of PRC2.1 function in Rb-deplete cells results in renewed palbociclib sensitivity. It is also important to understand the clinical implications of the results presented. The inclusion of these data would significantly improve the paper. However, besides some presentation issues and typos as described below, it is my opinion that the results are robust and of broad interest.Major questions:(1) Is the resistance to CDK4/6 inhibition conferred by mutation of MTF2 mediated by Rb?(2) Are mutations in PRC2.1 found in genetic analyses of tumour samples in patients with acquired resistance?

We thank the reviewer for their editing and experimental suggestions, and have integrated their responses into our re-submitted manuscript.

We also agree that understanding the role of RB1 in mediating palbociclib resistance to the proposed resistance mechanism is of particular interest. However, as there are three RB proteins expressed in human cells, this is a technically difficult question to probe genetically. Despite this technical challenge, we have provided multiple lines of evidence in our resubmitted manuscript that the resistance to palbociclib observed in our PRC2.1-deficent cells is mediated through the canonical CDK4/6-RB1 pathway. First, disruption of RB1 in HAP1 cells results in palbociclib resistance to a level comparable level to PRC2.1 disruption (Fig. 4E). Second, inactivation of SUZ12 or MTF2 increases the number of cells entering S-phase in palbociclib treatment (Fig. 4G) with no increase in basal rates of apoptosis (Fig. S2D), suggesting that any proliferation advantage observed in PRC2.1-defective cells is due to resistance to palbociclib-induced cell cycle arrest. Third, we show that over expression of CCND1 and CCND2 is sufficient to drive resistance to palbociclib in wild-type HAP1 cells (Fig. S5F). And finally, increased levels of CCND1 and CCND2 observed in cells lacking PRC2.1 activity results in higher CDK4/6 activity as measured by RB1 phosphorylation, despite palbociclib blockade (Fig. 6F). All these lines of evidence strongly suggest that MTF2-containing PRC2.1 regulates G1 progression in through the canonical CDK4/6RB1 pathway by repressing CCND1 and CCND2 expression.

Whether or not MTF2 deletion leads to palbociclib resistance in clinical samples is also of a question of particular interest. Currently, we are unaware of any reports that specifically mention MTF2 deletion as leading to palbociclib resistance, and we were unable to find another example in our own cancer database review. However, we have included references to other examples of MTF2 mutation resulting in chemotherapeutic resistance in our discussion. Additionally, although MTF2 is rarely observed to be mutated in cancers (Ngubo et al. 2023), it is highly differentially expressed and investigating decreased MTF2 transcription in palbociclib resistant tumors, though challenging, might prove fruitful. However, as mechanisms of palbociclib resistance is an area of active investigation, we speculate that future studies might uncover additional examples of MTF2 mediating resistance to this clinically important chemotherapeutic.

**Reviewer #2 (Public Review):**
Summary:Longhurst et al. assessed cell cycle regulators using a chemogenetic CRISPR-Cas9 screen in haploid human cell line HAP1. Besides known cell cycle regulators they identified the PRC2.1 subcomplex to be specifically involved in G1 progression, given that the absence of members of the complex makes the cells resistant to Palbociclib. They further showed that in HAP1 cells the PRC2.1, but not the PRC2.2 complex is important to repress the cyclins CCND1 and CCND2. This can explain the enhanced resistance to Palbociclib, a CDK4/6Inhibitor, after PRC2.1 deletion.Strengths:The initial CRISPR screen is very interesting because it uses three distinct chemicals that disturb the cell cycle at various stages. This screen mostly identified known cell cycle regulators, which demonstrates the validity of the approach. The results can be used as a resource for future research.The most interesting outcome of the experiment is the finding that knockouts of the PRC2.1 complex make the cell resistant to Palbociclib. In a further experiment, the authors focused on MTF2 and JARID2 as the main components of PRC2.1 and PRC2.2, respectively. Via extensive analyses, including genome-wide experiments, they confirmed that MTF2 is particularly important to repress the cyclins CCND1 and CCND2. The absence of MTF2 therefore leads to increased expression of these genes, sufficient to make the cell resistant to palociclib. This result will likely be of wide interest to the community.Weaknesses:The main weakness of the manuscript is that the experiments were performed in only one cell line. To draw more general conclusions, it would be essential to confirm some of the results in other cell lines.In addition, some of the findings, such as the results from the CRISPR screen as well as the stronger impact of the MTF2 KO on H3K27me3 and gene expression (compared to JARID2 KO), are not unexpected, given that similar results were already obtained before by other labs.

We thank the reviewer for their suggestions and we believe that we have addressed their main concern about the generality of the MTF2 regulation of D-type cyclin expression in our resubmitted manuscript. We have now shown through shRNA knockdown that MTF2 represses CCND1 in two additional cell lines, the breast cancer MDA-MB-231 and immortalized monkey COS7 cell line (Fig. 6E). However, it is important to note that MTF2 did not control CCND1 expression in every cell line tested (Fig. 6D), underscoring the context-dependent nature of this regulation. Future studies will illuminate what cell or tumor types in which this regulation is observed.

Additionally, while MTF2 has previously been shown to exert a greater effect on H3K27me3 levels in some circumstances (Loh et al. 2021, Rothberg et al. 2018), a number of notable reports in ES cell lines have concluded that PRC2 localization and H3K27me3 at the majority of genomic sites are dependent on both PRC2.1 and PRC2.2 activity (Healy et al. 2019, Højfeldt et al. 2019, Perino et al. 2020, Oksuz et al. 2018). Therefore, we think it is important to highlight the greater dependence on MTF2 for promoter proximal H3K27me3 levels in our transformed cell line context.

**Reviewer #3 (Public Review):**
This study begins with a chemogenetic screen to discover previously unrecognized regulators of the cell cycle. Using a CRISPR-Cas9 library in HAP1 cells and an assay that scores cell fitness, the authors identify genes that sensitize or desensitize cells to the presence of palbociclib, colchicine, and camptothecin. These three drugs inhibit proliferation through different mechanisms, and with each treatment, expected and unexpected pathways were found to affect drug sensitivity. The authors focus the rest of the experiments and analysis on the polycomb complex PRC2, as the deletion of several of its subunits in the screen conferred palbociclib resistance. The authors find that PRC2, specifically a complex dependent on the MTF2 subunit, methylates histone 3 lysine 27 (H3K27) in promoters of genes associated with various processes including cell-cycle control. Further experiments demonstrate that Cyclin D expression increases upon loss of PRC2 subunits, providing a potential mechanism for palbociclib resistance.The strengths of the paper are the design and execution of the chemogenetic screen, which provides a wealth of potentially useful information. The data convincingly demonstrate in the HAP1 cell line that the MTF2-PRC2 complex sustains the effects of palbociclib (Figure 4), methylates H3K27 in CpG-rich promoters (Figure 5), and represses Cyclin D expression (Figure 6). These results could be of great interest to those studying cell-cycle control, resistance mechanisms to therapeutic cell-cycle inhibitors, and chromatin regulation and gene expression.There are several weaknesses that limit the overall quality and potential impact of the study. First, none of the results from the colchicine and camptothecin screens (Figures 1 and 2) are experimentally validated, which lessens the rigor of those data and conclusions. Second, all experiments validating and further exploring results from the palbociclib screen are restricted to the Hap1 cell line, so the reproducibility and generality of the results are not established. While it is reasonable to perform the initial screen to generate hypotheses in the Hap1 line, other cancer and non-transformed lines should be used to test further the validity of conclusions from data in Figures 4-6. Third, conclusions drawn from data in Figures 3D and 4D are not fully supported by the experimental design or results. Finally, there have been other similar chemogenetic screens performed with palbociclib, most notably the study described by Chaikovsky et al. (PMID: 33854239). Results here should be compared and contrasted to other similar studies.

We thank the reviewer for their suggestions regarding our manuscript. While the genes recovered as mediating cellular responses to camptothecin and colchicine was never confirmed following our chemogenetic screens, we felt our primary findings were in the area of palbociclib resistance and decided focus our follow-up investigations on genes. We included the results camptothecin and colchicine chemogenetic screens as confirmation of the specificity of PRC2 mutation resulting in resistance to palbociclib (Fig. 4C) and for others in the community to use as a resource for future investigations. We have also clarified our results for Figure 3D and 4D in our revised manuscript, as well as included additional plots of these results (Fig. S1DS1F). And, with our resubmitted manuscript, we believe we have addressed their concern of the generality of our results by demonstrating our primary finding that MTF2 regulates D-type cyclins in additional cell lines other than HAP1. We feel these results indicate that while not “general”, there are additional cellular contexts that our main result holds true. In line with this, and to address how our chemogenetic screens fits into the landscape of previous studies, including Chaikosvsky et al., we have included the following lines to our discussion: “Additionally, other chemogenetic screens utilizing palbociclib and have not identified that inactivation of PRC2 components as either enhancing or reducing palbociclib-induced proliferation defects, suggesting that PRC2 mutation is neutral in the cell lines studied. These observations not only underscore the context-dependent ramifications of mutation of these PRC2 complex members, but also may help inform the context in which CDK4/6 inhibitors are most efficacious.”

**Recommendations for the authors:**

**Reviewer #1 (Recommendations For The Authors):**
(1) "We found that only thirteen and twenty genes resulted in sensitivity or resistance, respectively, in every conditions tested and were deemed non-specific and excluded from any further analysis (see Table S2)." It's unclear to me why these genes were deemed 'nonspecific'. Are these genes functionally important for the general exclusion of xenobiotic molecules?

By this, we simply meant that these effects were not specific to one condition. Such genes could affect drug half-life or a general stress response, but are less likely to have functions directly tied to the pathway targeted by a drug than are genes whose loss affects only one condition.

(2) "Given that increased CCND1 levels is sufficient to drive increased CDK4/6 kinase activity, upregulation of these D-type cyclins is likely to be a significant contributor to the palbociclib resistance in MTF2∆ cells." It's unclear to me what is the basis for this statement. This is only true if there is free CDK4/6. If CDK4/6 is already fully occupied by D-type cyclins, then increased CCND1 levels would not be expected to have an effect.

While we anticipated that increased levels of CCND1 would result in more CDK4/6-Dtype association, we now demonstrate in the new Figure S5F that there is more CCND1 in complex with CDK6 in both SUZ12∆ and MTF2∆ cell lines. Furthermore, we able to show in Figure S5G that overexpression of D-type cyclins results in resistant to palbociclib-induced proliferation defects in HAP1 cells.

(3) The description of the results is very confusing in places, especially regarding "resistance" versus "sensitivity" genes. For example: "CCNE1, CDK6, CDK2, CCND2 and CCND1, all of which are integral to promoting the G1/S phase transition, ranked as the 2nd, 24th, 27th, 29th and 46th most important genes for palbociclib resistance, respectively (Figures 1F and 1G). CCND1 and CCND2 bind either CDK4 or CDK6, the molecular targets of palbociclib, whereas CDK2 and CCNE1 form a related CDK kinase that promotes the G1/S transition.Similarly, cells with sgRNAs targeting RB1, whose phosphorylation by CDK4/6 is a critical step in G1 progression, displayed substantial resistance to palbociclib." My reading of this paragraph suggests that disruption of the CDK6 locus is associated with palbociclib resistance - surely this is a typo and instead should have been sensitivity? Please explain.

We thank the reviewer for pointing this out and have corrected this typo

(4) Sensitivity to palbociclib was enhanced in cells expressing sgRNAs targeting H4 acetylation, positive regulators of Pol II transcription, and regulators of the DNA Damage Response pathway (Figures 3A and 3B), although this sensitivity was much weaker than that seen with DNA damaging agents. This observation is consistent with long-term treatment with palbociclib inducing DNA damage, as has been suggested by a number of recent publications 65,66." This is also consistent with recent work on Cdk7 inhibitors (Wilson et al. Mol Cell 2023), as Cdk7 inhibition is expected to affect both CDK1/2/4/6 activities and Pol II transcription.

We thank the reviewer for bringing this observation to our attention and we have added this citation to this passage in our manuscript.

(5) Figure 3D - would it not make sense to plot the data such that palbo concentration is on the x-axis? It is also difficult to interpret since the data are normalized to starting "% proliferation" at the indicated palbo treatment, when it is likely that % proliferation changes significantly with palbo concentration. Indeed, this is the graphing format used for a later figure (Figure 4D). The data with rotenone suggests palbo antagonizes rotenone-mediated reduction in proliferation. But it's unclear to me whether the graph shows the converse - that rotenone treatment modulates palbo-induced cell cycle arrest.

This reviewer is correct about the fact that increasing doses of palbociclib in the absence of oxidative phosphorylation do indeed have an effect on proliferation. However, it is helpful to normalize proliferation values to each initial dose of palbociclib and then compare this to the different oxidative phosphorylation inhibitors treatment combinations. To illustrate that the oxidative phosphorylation inhibitors do indeed antagonize palbociclib-induced proliferation defects, we have now included the data graphed as each oxidative phosphorylation inhibitor vs palbociclib as Supplemental Figures S1D-S1F.

• The highest concentration of GSK126 tested (5µM) does not appear to confer resistance, but perhaps this is due to off-target effects or cytotoxicity?

We agree with the reviewer that at the highest doses of dose of GSK126, low doses of palbociclib do not confer resistance to palbociclib. However, higher doses do appear to have this effect. We have included a statement in our results section to address this reviewer’s observations.

• Disruption of Emi1 leads to resistance (Figure 1F, FZR1), yet overexpression induces resistance (Mouery et al. bioRxiv 2023). Explain.

We do not understand why EMI1 responds in this way, and therefore we cannot comment on this in the text.

Typos/stylistic comments:• Typo "However, the net result of these opposing effects on cell cycle progression, and the contribution of the individual subcomplexes to this regulation, rained unclear."

We thank the reviewer for pointing this out, and we have corrected it.

• Use of the word "growth" - I think the authors should be more precise. Is "proliferation" meant here?

We thank the reviewer for pointing this out, and we have corrected it.

• n Figure 4G, two of the panels have 8.42%. Is this correct, or may it be a copy/paste error?

This was an error, but is no longer relevant as we have reconducted and reanalyzed this experiment.

Reviewer #2 (Recommendations For The Authors):Major Points(1) Some of the conclusions should be confirmed in additional cell lines. I would suggest testing the resistance to Palbociclib in several additional cell lines, where MTF2 and JARID2 are deleted. If the conclusion can be generalized, one would expect that the differential role of MTF2 versus JARID2 can be confirmed in more cell lines.

While the PRC2.1-dependent repression of D-type cyclins does not appear to be general, we have now demonstrated in Figures 5SE and 6F that there are multiple different cellular contexts in which our observations are consistent. Specifically, we demonstrate that GSK126 causes upregulation of CCND1 in both immortalized nontumor cells (COS7 cells) and in the breast cancer cell line MDA-MB-231. Moreover, in both cases we showed that this effect is PRC2.1-dependent, as shRNA knockdown of MTF2 increases expression of CCND1.

(2) In addition, it may be attractive to make use of publicly available RNA-seq data of MTF2 and JARID2 knockout/down cells, to investigate the generality of the finding that PRC2.1 regulates CCND1 and CCND2.

While it would be useful to address this issue, Figure S5E demonstrates that the repression of D-type cyclin expression by PRC2.1 is context dependent. Furthermore, prior to identifying the lines shown in Figure 6F and 5SE, we were not aware of which lines to focus our investigations on. However, we have now demonstrated a few cellular contexts in which either chemical inhibition of PRC2 or knockdown of MTF2 results in de-repression of CCND1 expression.

(3) At a bare minimum the authors should strongly discuss the limitations of the study, and tone down the conclusions.

We would agree with this based upon the data in the original submitted manuscript, however, now that we have shown that this effect is more general, this is less critical. That said, we do not see this effect in all cell lines, and we have made this apparent in the final version of the manuscript.

Minor point(1) In my view, Figures 1-3 should be shortened to the most essential points, and some data/figures should be moved to the supplementary figures. Especially the STING genenetwork graphs are in my view not particularly meaningful.

While we understand the opinion of this reviewer, we feel that these data will be of significant interest to some readers.

(2) Figure 6E and 6F/G appear to be largely redundant. This can perhaps be made more concise.

This has been addressed in the new version of Figure 6

(3) Figure 5D should be enlarged.

We thank the reviewer for this suggestion and have enlarged the image.

**Reviewer #3 (Recommendations For The Authors):**
The manuscript could be edited to improve clarity. In several places, the scientific logic motivating an experiment is confusing, and there are several hypotheses and conclusions that seem opposite from what the data are suggesting. Some aspects of the figures were also unclear. Specific examples include the following:(1) Last sentence of abstract : "Our results demonstrate a role for PRC2.1, but not PRC2.2, in promoting G1 progression." Data show that knockout of PRC2.1 components promotes G1 progression through upregulation of CycD, so the conclusion here is the opposite.

We thank the reviewer for catching this error. We have now changed this to “in antagonizing G1 progression”.

(2) In the second paragraph of the results, CCNE1, CDK2, etc are described as scoring high for palbociclib resistance, but those genes scored as sensitizing. Also, in that paragraph, it is described that a drug is sensitizing cells to loss of a gene, which seems like incorrect logic. It should be clarified that knock-out of a gene either sensitizes or desensitizes cells to the drug.

We thank the reviewer for catching this error. We have now corrected it.

(3) In the motivation for the experiment in Figure 3D, it is written: "we asked whether chemical inhibition of oxidative phosphorylation could rescue sensitivity to palbociclib". Considering that knock-out of genes that mediate oxidative phosphorylation confer resistance to palbociclib, it is confusing why it was expected that chemical inhibitors would restore sensitivity.

We are sorry if the original wording was confusing. We have now changed this to “combined inhibition of oxidative phosphorylation and CDK4/6 activity mutually rescue the proliferation defect imposed by agents targeting the other process”.

(4) If the intention of Figure 3D is to test the hypothesis that chemical inhibition of oxidative phosphorylation modulates sensitivity to palbociclib, the clarity of Figure 3D would be improved if data were shown such that palbociclib concentration is on the x-axis and the different curves are different drug concentrations.

It appears that there is some mutual suppression, which inhibition of each process rescues cells partly from inhibition of the other. In fact, with these drugs the stronger of the two is seen as the rescue of mitochondrial poisons by palbociclib. We have now discussed this in the text.

(5) The authors should check the units on the x-axis in Figure 4D, should they be log[uM Palbo] or log [nM Palbo]?

We thank the reviewer for catching this error. We have now corrected it

(6) It should be clarified which data are summarized in the graph to the right in Figure 4G, are these experiments with palbociclib?

This is currently included in the figure legends.

(7) The text suggests that the control CCNE1 knockout is shown in Figure 4E, but those data are missing.

This has been corrected in Figure 4E.

Several conclusions are not well supported by the data and should be revised or more data and analysis should be added.(1) The titular conclusion that the "PRC2.1 Subcomplex Opposes G1 Progression through Regulation of CCND1 and CCND2" has only been demonstrated in the context of a Cdk4/6 inhibitor in HAP1 cells. There is little evidence supporting this claim that is broadly applicable. For example, data in Figure 4G show small and not demonstrable significant differences in G1 and S phase populations in the mock experiments. Also, experiments in other cells are needed to support the rigor and generality of the conclusion.

Our chemogenetic screen and competitive proliferation assay data in Figure 4A, 4C and 4E support the conclusion that PRC2.1 and PRC2.2 play opposing roles in G1 progression. Furthermore, we have repeated the initial BrdU incorporation experiments shown in Figure 4G and have been able to demonstrate that JARID2∆ cells do indeed display a significant decrease of cells entering into S-phase when treated with palbociclib. Most importantly, in the Figures 6D and 6E we show additional cell lines where this is the case. Therefore, we feel that this title is valid in the current version of the manuscript, where we have shown it to be the case in multiple tumor-derived human cell lines as well as immortalized non-human primate cells.

(2) It is unclear how the data in Figure 3D support the conclusion that the administered inhibitors of oxidative phosphorylation influence response to palbociclib.

As noted in the response to point 4, we have now discussed this mutual rescue more thoroughly in the text.

(3) In Figure 4D, the IC50 values should be calculated and statistical significance based on biological replicates should be determined. Also, the conclusion that "increasing doses of GSK126 withstood palbociclib-induced growth suppression" is overstated, as ultimately all drug conditions succumb to palbocilib suppression of proliferation, although there may be differences in sensitivity.

We have now included a statical analysis of each data point in Figure 4D.

**Editorial comments:**
(1) The title does not seem to optimally capture the content of the paper. Please consider changing it, e.g. focusing on palbociclib resistance.

While we used this particular drug to make the original observation, we feel it is more general to discuss the underlying biology (cyclin gene control) than the pharmacological methodology. Moreover, we have now extended our findings about the regulation of D-type cyclins by PRC2.1 to several cell lines, derived from both cancers and primary cells, re-enforcing the fact that this effect is observed more broadly.

(2) Please indicate the biological system (haploid human HAP1 cells) in either title or abstract.

The abstract now indicates that we have observed this in CML, breast cancer and immortalized primary cells.